# SKILL WEAVING: EFFICIENT SELF-IMPROVEMENT OF LLMS VIA MODULAR SKILLPACKS

## ABSTRACT

In this work, we introduce SkillWeave, a modular self-improvement framework that enables large language models to specialize using only their own generations. SkillWeave partitions full capabilities of a general-purpose model into domain-specific skillpacks—lightweight, domain-specific delta modules—that reorganize and refine the model's internal knowledge. To ensure deployment efficiency, we further propose SkillZip, a fully quantized delta-compression method that eliminates runtime decompression and achieves low-latency inference. By combining structured specialization with hardware-aware design, SkillWeave delivers both parameter-efficient specialization and inference-efficient execution. On multi-task and agentic benchmarks, a 9B SkillWeave model outperforms task-specific baselines and even surpasses a 32B monolithic LLM, while achieving up to 4× speedup.

## 1 INTRODUCTION

Large Language Models (LLMs), pretrained on diverse corpora, exhibit impressive generalization via post-training methods like supervised fine-tuning (SFT). However, such methods require increasingly expensive and less informative human-crafted data as LLMs improve (Zhang et al., 2025; Gan & Liu, 2025), while the growing diversity of downstream domains further exacerbates this data bottleneck (Jang et al., 2025; Lee et al., 2025). A natural alternative is employ even larger LMs, but this in turn imposes prohibitive burdens on memory and inference latency (Ping et al., 2024). This raises a central question: *how can we enable LLMs to sustain multi-domain performance under fixed memory and latency budgets, without relying on external labels or stronger teacher models?*

To tackle this challenges, We propose `SkillWeave`, an efficient self-improvement framework that enables an LLM to enhance itself under fixed memory and inference budgets. The model partitions its full capability space into multiple domains and learns domain-specialized skillpacks on each domin through self-generated supervision. Each skillpack is then compressed into an lightweight and inference-efficient format to support scalable storage and deployment.

**The central innovation of `SkillWeave`** is to decompose the model's monolithic capabilities into a diverse set of domain-specific skillpacks, then compact them back into deployable form. Our design addresses two fundamental limitations overlooked in prior work, both centered on the underappreciated role of LLM's internal capability structure. **First.** Full-parameter fine-tuning enables high capacity but incurs prohibitive storage and inference overhead, while PEFT (Ding et al., 2023) methods (*e.g.,* LoRA (Hu et al., 2022)) is lightweight but often underperform and fail to retain sufficient ability of specific domain. `SkillWeave` resolves this trade-off by first training each skillpack with the full parameter budget to absorb fine-grained ability, followed by compact compression for deployment. **Second.** Monolithic LLMs entangle all domains within a single shared parameter space, while such space sharing imposes a bottleneck due to task interference, leading to suboptimal performance and catastrophic forgetting (Yadav et al., 2024; Du et al., 2024). In contrast, `SkillWeave` decompose the LLM's overall capacity into skillpacks and isolates training by domain, producing independent skillpacks that preserve self-refined performance on each domain while avoiding cross-task interference.

A key step of `SkillWeave` is its selective use of self-generated data for skillpack training. Unlike prior self-improving methods (Kang et al., 2024b;a) that treat all synthetic data as equally reliable, we incorporate lightweight *rule-based verification* mechanism to automatically filter helpful and harmful samples. We then apply Direct Preference Optimization (DPO) (Rafailov et al., 2023) to reinforce

reliable behaviors while suppressing erroneous patterns. This targeted curation strategy enhances the stability and quality of self-distillation with minimal resources.

While domain skillpacks enable fine-grained specialization, naïvely retaining multiple full-parameter skillpacks is impractical due to storage and inference costs. To close this gap, **we propose `SKillZip`, a novel and inference-efficient compression strategy** tailored for modular deployment. By first merging shared knowledge into a common backbone, `SKillZip` disentangles task-specific skill-packs and reinforces core capabilities. At its core, it employs a fully quantized design that compresses both weights and activations, eliminating the need for runtime decompression, re-expansion, or costly dequantization. This design significantly reduces inference latency, contrasts with prior delta-compression methods (Yuan et al., 2023; Ping et al., 2024) that primarily reduce storage size. Furthermore, we integrate hardware-aware smoothing techniques to support low-bit quantization without performance degradation. In this way, SkillZip preserves the benefits of full-parameter specialization while achieving low-latency, resource-efficient inference, completing the SkillWeaving pipeline from self-improvement to deployment.

In summary, `SkillWeave` provides a unified pipeline for modular and efficient self-improvement. By decomposing model capabilities into compact skillpacks, `SkillWeave` reorganize and refine the model's internal knowledge structure while ensuring parameter and inference efficiency with minimal overhead.

This paper makes three significant contributions:

- We propose `SkillWeave`, a modular self-improvement framework that enables an LLM to enhance itself under fixed memory and inference budgets. By partitioning capabilities into domain-specific skillpacks and integrating them through a shared backbone, `SkillWeave` achieves scalable and interpretable self-improvement.

- We introduce `SKillZip`, a fully quantized delta compression strategy tailored for modular deployment. Unlike prior approaches focused on weight storage, `SKillZip` jointly quantizes weights and activations, eliminating runtime decompression and delivering low-latency inference with hardware-aware optimization.

- We validate our method on multi-task and agentic benchmarks, where a 9B SkillWeave model outperforms task-specific models and a 32B monolithic model, while achieving 4× faster inference speedup and superior fidelity compared to existing delta-compression baselines.

## 2 RELATED WORK

### 2.1 SELF IMPROVEMENT VIA SYNTHETIC DATA

Self-improvement methods seek to enhance large language models (LLMs) using their own synthetic generations, thereby avoiding human annotations and external teachers. Self-Specialization (Kang et al., 2024b) fine-tunes an LLM on its self-produced data to induce task-specific expertise, but the absence of supervision or filtering often results in unstable and suboptimal behaviors. Self-MoE (Kang et al., 2024a) extends this paradigm by routing tasks to independently self-specialized LoRA experts, offering modularity but still constrained by the limited capacity of lightweight adapters.

Another line of work leverages LLM-as-a-judge prompting to annotate synthetic data, followed by preference-based optimization such as DPO (Rafailov et al., 2023). Representative examples include Self-Rewarding (Yuan et al., 2024) and Meta-Rewarding (Wu et al., 2024), while Self-Align (Sun et al., 2023) extends this paradigm by guiding LLMs with human-written principles that trigger matching rules. Similarly, RLAIF (Lee et al., 2024) reuses the RLHF pipeline but relies on synthetic judgments in place of human feedback. Although these methods enhance robustness, they are often hindered by unreliable judgments, reward bias, and expensive training overhead.

### 2.2 TASK VECTOR MERGING AND COMPRESSION

Task Arithmetic first (Ilharco et al., 2023) introduced the concept of task vectors - defined as the difference between a fine-tuned model and its base. Subsequent works (*e.g.,* Ties-Merging (Yadav

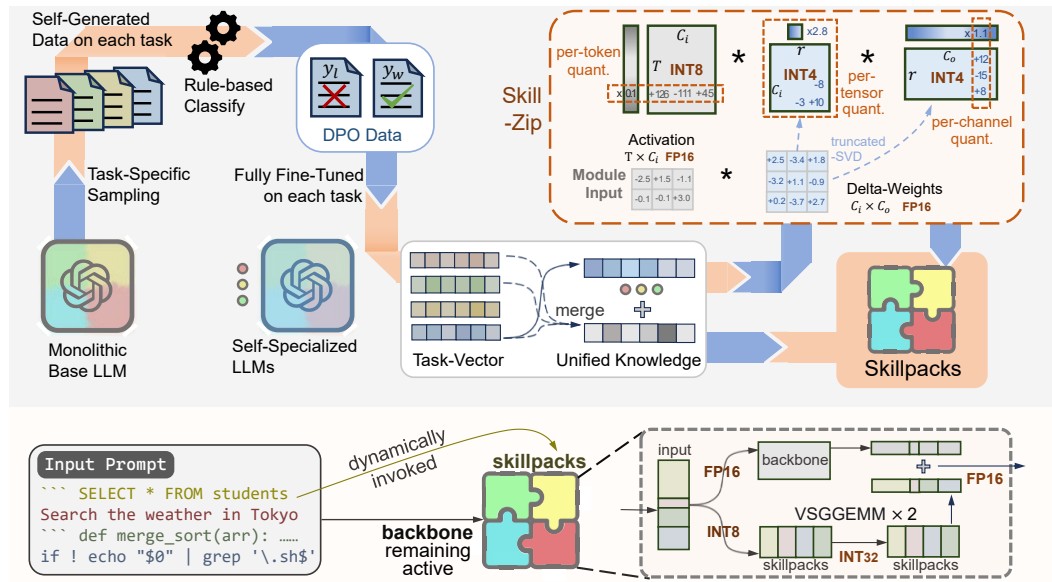

Figure 1: Overview of the `SkillWeave` framework. The **top** section illustrates the full pipeline, consisting of two stages: (1) decomposing a monolithic language model into task-specific skill vectors via preference-based training, and (2) compressing each skillpack into an inference-friendly form using structured quantization. The **bottom** section demonstrates the integration and inference process under an agent-style serving scenario. As a whole, the figure shows how `SkillWeave` reorganizes and refines the model's internal knowledge structure to balance performance gains and parameter efficiency.

et al., 2024), DARE (Yu et al., 2023) and PCB-Merging (Du et al., 2024)) applied it to merging large language models.

Meanwhile, recent works have explored delta-compression to reduce the overhead of storing and serving multiple task vectors. BitDelta (Liu et al., 2024a) introduces 1-bit quantization for delta weights with scaling factors, significantly reducing storage. SVD-based methods, such as SVD-LLM (Wang et al., 2024c), ASVD (Yuan et al., 2023) and Twin-Merging (Lu et al., 2024), leverage low-rank decomposition to compress task-specific deltas (Ryu et al., 2023; Gargiulo et al., 2024). Yet, these method are reported to yield suboptimal accuracy. DeltaCome (Ping et al., 2024) and GPT-Zip (Isik et al., 2023) further combine quantization with sparsification, but sparsity offers limited benefit under current hardware constraints during inference.

## 3 METHODOLOGY

In this section, we introduce `SkillWeave`, a modular self-alignment framework that decomposes monolithic language models into compact, self-specialized *skillpacks*.

**Problem setting.** We assume access to a base model $\theta_0$ and a small seed dataset $\mathcal{D}_{\text{seed}}$ covering the model's core capabilities. Our objective is to construct a collection of lightweight modules $\{\mathcal{S}_1, \ldots, \mathcal{S}_K\}$, each targeting a distinct capability (e.g., mathematics, coding, reasoning, dialogue). Each skillpack should satisfy three criteria: (i) it accurately captures domain-specific improvement learned from self-generated data, (ii) it preserves general competence by avoiding interference across domains, and (iii) it can be deployed efficiently alongside other skillpacks on modern inference engines.

**Overview of the SkillWeave pipeline.** SkillWeave consists of three stages, forming the central methodological contribution of this work:

1. **Skillpack Building (Self-Specialization).** We isolate domain signals by independently training the base model on self-generated preference data for each capability. This produces

a collection of *task vectors* capturing domain-specific improvement, while avoiding severe cross-domain conflicts.

2. **Skillpack Compression (Full-tuning–then-zip).** We first unify shared knowledge through model merging and then compress each skill vector into a compact, inference-friendly representation. This compression is suitable for high-throughput serving, while retaining the domain-specific knowledge in *skillpacks*.

3. **Skillpack Integration (Modular Deployment).** At inference time, the shared backbone $\theta_0$ remains always active, while exactly one skillpack is selected based on the input domain. The selected module enabling multiple capabilities to coexist in memory and achieving scalability.

## 3.1 Skillpack Building

The goal of this stage is to extract a set of task-specific, self-improving vectors—referred to as *proto-skillpacks*—from a base LLM through preference-driven finetuning. This is achieved via a self-alignment pipeline that is fully automated and requires no external human supervision.

**Task Decomposition.** We begin by partitioning a seed instruction dataset $\mathcal{D}_{\text{seed}}$ into $K$ disjoint subsets $\mathcal{D}_1, ..., \mathcal{D}_K$, each associated with a specific task $\mathcal{T}_k$ such as dialogue, reasoning, or web search. Each subset serves as the anchor for self-supervised alignment on its respective capability.

**Self-Generation.** For each task $\mathcal{T}_k$, we query the base LLM $\mathcal{M}_0$ using the prompts in $\mathcal{D}_k$ to produce candidate outputs. This results in a self-generated dataset $\mathcal{D}_k^{\text{gen}} = \{(x_i, y_i^{\text{gen}})\}$ containing model completions for the task-specific instructions. The generated responses may vary significantly in quality, containing both valuable and potentially harmful outputs.

**Rule-Based Classification.** To enable effective learning from these mixed-quality generations, we apply a lightweight rule-based filter that separates $\mathcal{D}_k^{\text{gen}}$ into two preference-ranked subsets: $\mathcal{D}_k^+$ = Helpful samples, $\mathcal{D}_k^-$ = Harmful samples. These rules are defined at the task level and capture high-level failure patterns, such as instruction misalignment in dialogue, contradiction in reasoning or test error in coding. This classification is interpretable, using minimal resources, while providing consistent and stable supervision for alignment.

**Preference Optimization.** Given the preference-labeled dataset $(\mathcal{D}_k^+, \mathcal{D}_k^-)$, we fine-tune the base LLM using online Direct Preference Optimization (DPO) (Rafailov et al., 2023). This approach minimizes a contrastive objective that encourages the model to prefer helpful over harmful outputs:

$$\mathcal{L}_{\text{DPO}}(\pi_\theta; \pi_{\text{ref}}) = -\mathbb{E}_{(x, y_w, y_l) \sim \mathcal{D}}[\log \sigma(\beta \log \frac{\pi_\theta(y_w \mid x)}{\pi_{\text{ref}}(y_w \mid x)} - \beta \log \frac{\pi_\theta(y_l \mid x)}{\pi_{\text{ref}}(y_l \mid x)})]. \quad (1)$$

This yields a new model $\mathcal{M}_k = \mathcal{M}_0 + \Delta_k$, where $\Delta_k$ is the *task vector* containing the self-refined, task-specialized knowledge for task $\mathcal{T}_k$. We treat each task vector $\Delta_k$ as a *proto-skillpack* —serving as modular carriers of task-specific skills.

## 3.2 Skillpacks Compression

**From Task Vectors to Skillpacks.** While these full-parameter vectors enable high-quality refinement and task-specific adaptation, naively retaining them imposes severe costs in storage and inference. To make modular specialization feasible in practice, we propose `SKillZip`, a inference-efficient compression strategy that introduces *full-quantization for delta-compression*.

**Model Merging for Shared Knowledge.** Before compression, we extract shared cross-task knowledge across task-specific deltas to isolate task-specific skills. Concretely, we compute a shared component through model-merging: $\Delta_{\text{shared}} = \text{Merge}([\Delta_1, \ldots, \Delta_k])$, integrate them back into the model's backbone and subtract it from each delta: $\Delta_i \leftarrow \Delta_i - \Delta_{\text{shared}}$. This enhances the backbone with generalized knowledge while making individual deltas sparser and more task-specific, facilitating subsequent compression.

**Full Quantization for Delta-Compression.** Existing delta-compression methods compress only the parameter weight, and during inference, these compressed deltas must first be dequantized to floating-point (e.g., FP16). In contrast, SKillZip is the first to apply *full-quantization* to delta-compression—quantizing both the deltas and the corresponding activation inputs. This enables direct computation in low-bit integer formats (e.g., INT8 or INT4) using hardware-accelerated tensor cores, eliminating the need for runtime reconstruction. As a result, SKillZip achieves significantly higher inference speed while maintaining compression fidelity.

Concretely, for a linear delta $W$ and input activations $X$, we seek a $k$-bit static quantizer $\text{Quant}_k(\cdot, q)$ and low-rank factors $A \in \mathbb{R}^{C_i \times R}, B \in \mathbb{R}^{R \times C_o}$ to be quantified (s.t. $\text{diag}(\vec{s}_B) \cdot \hat{B} \approx B$), whose quantized forms $\hat{X}, \hat{A}, \hat{B}$ minimize the end-to-end reconstruction error:

$$\hat{A}, \hat{B}, q = \underset{\hat{A}\ \hat{B}\ q}{\arg\min} \|XW - \hat{X}\hat{A}\hat{B}\|, \quad \hat{X} = \text{Quant}_k(X, q), \ \hat{A} = \text{Quant}_k(A), \ \hat{B} = \text{Quant}_k(B). \quad (2)$$

In hardware-accelerated GEMM (Lawson et al., 1979) kernels, FP16 scaling can only be efficiently applied along the outer dimensions—that is, before and after the INT8 matrix multiplication. Scaling between intermediate steps (*i.e.,* $\hat{A} \cdot \text{diag}(\vec{s}_A) \cdot \text{diag}(\vec{s}_B) \cdot \hat{B}$) is constrained by hardware. Hence we are constrained to adopt per-token/per-tensor quantization for $\hat{X}$, per-tensor for $\hat{A}$, and per-tensor or per-channel for $\hat{B}$ only. Let $\vec{s}_X, \vec{s}_B$ be per-token/per-channel scaling vector for quantized matrix $\hat{X}, \hat{B}$, $s_A$ be a per-tensor scale fo $\hat{A}$, the reconstructed output is then computed by:

$$Y = XAB \approx \text{diag}(\vec{s}_X) \cdot \hat{X} \cdot s_A \cdot \hat{A} \cdot \hat{B} \cdot \text{diag}(\vec{s}_B) = \text{diag}(s_A \cdot \vec{s}_X) \cdot \hat{X}\hat{A}\hat{B} \cdot \text{diag}(\vec{s}_B). \quad (3)$$

**Double Smoothing for Quantization Fidelity.** In the quantization field, accuracy degradation often arises due to outliers — a small number of activation channels whose magnitudes are 100× larger than the rest. When misaligned with quantization axes, these outliers inflate the dynamic range, forcing most values in each quantized block to collapse to zero, thereby leading to significant quantization error.

First, we apply **channel-wise smoothing** to reduce outliers in the activation channels. As shown in Fig. 2, activation outlier channels display task-specific patterns that remain remarkably stable across inputs within the same domain. Inspired by SmoothQuant (Xiao et al., 2023), We learn a task-specific scale vector $s \in \mathbb{R}^{C_i}$ that rebalances the activation and weight magnitudes: $Y = (X \cdot \text{diag}(1/s)) \cdot (\text{diag}(s) \cdot W) = X_{\text{smooth}} \cdot W_{\text{smooth}}$. This transformation shifts some of the quantization burden from activation to weight, enabling both to be quantized more uniformly.

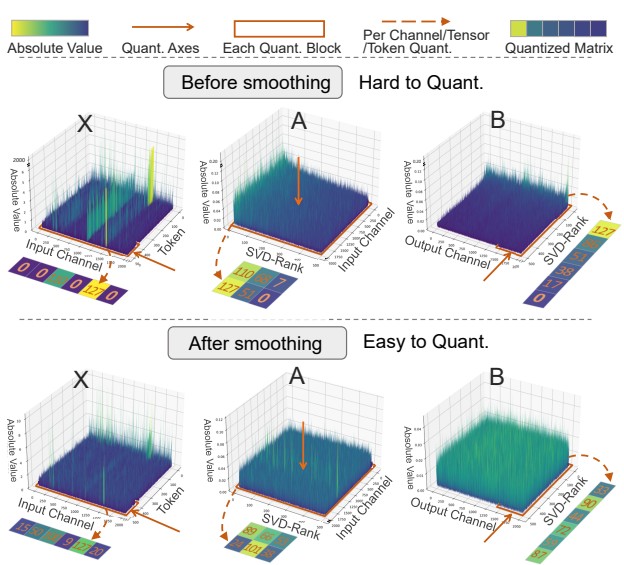

Figure 2: The activation $X$ and the low-rank weight matrices $A$ and $B$ are initially hard to quantize due to outliers misaligned with quantization axes. By normalizing channel scales of $X$ and spreading singular value energy across $A$ and $B$, the resulting matrices become significantly more quantization-friendly.

Next, we apply truncated singular value decomposition (SVD) to the smoothed weight matrix $W_{\text{smooth}}$: $W_{\text{smooth}} \approx U_R \Sigma_R V_R^T$, $A = U_R \Sigma_R^{1/2}$, $B = \Sigma_R^{1/2} V_R^T$. However, the concentration of energy in top singular vectors (*i.e.,* the first few columns in $A$ and rows in $B$) leads to new outliers misaligned with quantization axes (per-tensor/per-channel). To alleviate this, we apply a second-stage **rank-wise smoothing**: a appropriate orthogonal rotation matrix $Q$ to disperses the energy evenly across dimensions: $A_{\text{rot}} = AQ$, $B_{\text{rot}} = Q^T B$. Together, as demonstrated in Fig. 2, these two smoothing steps significantly reduce quantization distortion without loss of information.

### 3.3 SKILLPACK INTEGRATION

At inference, our architecture maintains a unified yet modular computation flow. The shared backbone of the model remains constantly active, ensuring a stable foundation across tasks. In parallel, task-specific skillpacks are dynamically invoked based on the task type inferred from each input.

Concretely, each Transformer block is equipped with shared weights $W$ and multiple task-specific low-rank matrices $\hat{A}_i, \hat{B}_i$. Given input activations $X$, the shared computation $XW$ proceeds as usual—either in FP16 or quantized precision. Meanwhile, tokens are grouped by task type into $\{X_1, X_2, \ldots, X_K\}$, and each group is processed with its corresponding skillpack:

$$\text{Output}_i = X_i W + \hat{X}_i \hat{A}_i \hat{B}_i. \tag{4}$$

## 4 EXPERIMENTAL SETUP

### 4.1 BASELINE METHODS

We compare `SkillWeave` against a comprehensive set of baselines spanning both **single-skill** and **multi-skill** self-improvement settings.

**Single-Skill Baselines.** In the single-skill setting, we evaluate each skillpack independently on its corresponding domain. We consider two categories of baselines:

- **Alternative *parameter formats* baselines.** 1) PEFT: Replace our full-parameter-tuning-then-SkillZip pipeline with LoRA fine-tuning. 2) Full-Parameter Finetuning. 3) Delta-Compression: State-of-the-art delta-compression methods, such as BitDelta (Liu et al., 2024a), SVD-based methods (Wang et al., 2024c; Yuan et al., 2023), and DeltaCome (Ping et al., 2024).
- **Alternative *self-improvement algorithms*.** 4) Self-Specialization (Kang et al., 2024b). 5) Self-Rewarding based methods: Self-rewarding (Yuan et al., 2024), Self-Align (Sun et al., 2023).

**Multi-Skill Baselines.** The multi-skill setting is the primary application scenario for `SkillWeave`, where all skillpacks are simultaneously evaluated across diverse tasks. We consider a broad range of competitive baselines:

6) Open-Source LLMs. 7) Multi-Teacher Distillation: FuseLLM (Wan et al., 2024b) and FuseChat3.0 (Yang et al., 2025), as a upper bound. 8) Routing-Based Methods: Self-MoE (Kang et al., 2024a), LoRA-MoE (Gao et al., 2024) and Twin-Merging (Lu et al., 2024), the most relevant methods to ours. 9) Multi-Task Learning. 10) Model Merging and Model Grafting: Task Arithmetic (Ilharco et al., 2023), Ties-Merging (Yadav et al., 2024) and others.

Although baselines (8)–(10) were not originally designed for self-improvement, we adapt them accordingly to ensure fair comparison. Each relies only on self-generated data and self-driven optimization, consistent with our setting. Together, these baselines provide a meaningful reference for evaluating modular specialization in multi-task environments.

### 4.2 EVALUATION SCENARIOS AND BENCHMARKS

We design experiments across two key evaluation scenarios: (1) a general-purpose **multi-capacity** setting, and (2) a practical **LLM-as-Agent** deployment setting. These reflect both academic and real-world uses of modular, skill-based language models.

**General Capabilities Evaluation.** In this setting, we define four core tasks representative of general LLM capabilities: *dialogue*, *reasoning*, *math*, and *coding*. Each domain is evaluated using multiple established benchmarks. These tasks cover a broad spectrum of instruction-following abilities, ensuring that our evaluation is both comprehensive and robust.

**LLM-as-Agent Evaluation.** To evaluate this setting, we adopt **AgentBench** (Liu et al., 2024b), and focus on 5 diverse and representative domains of AgentBench: *Database (DB)*, *Operating System (OS)*, *Knowledge Graph (KG)*, *Web Shopping (WS)*, and *Web Browsing (WB)*. These tasks are tailored to assess LLMs acting as autonomous agents across a wide range of environments.

Table 1: Overall performance of `SkillWeave` and multi-skill baselines in the general capability setting using Llama-3.1-8B-Instruct as the backbone. The best results for baselines are shown in **bold**, our results are highlighted with a pink background, and performance gaps between the best and `SkillWeave` are marked in **green**.

| Method | #Params | Mathematics | | Coding | | Dialogue | | Reasoning | |
|---|---|---|---|---|---|---|---|---|---|
| | | GSM8k | MATH | HumanEval | MBPP | AlpacaEval2 | IFEval | BBH | ARC-C |
| **Open-Source LLMs** | | | | | | | | | |
| Llama3.1-8B-Instruct | 8B | 84.5 | 51.9 | 69.5 | 75.4 | 28.3 | 75.9 | 65.8 | 82.4 |
| Qwen1.5-72B-Chat | 72B | 82.7 | 42.5 | 71.3 | 71.9 | 40.6 | 77.1 | 68.3 | 75.8 |
| Gemma2-27B-it | 27B | 90.4 | 54.4 | 78.7 | 81.0 | 58.9 | 77.1 | 74.9 | 77.4 |
| Qwen2.5-14B | 14B | 90.2 | 55.6 | 56.7 | 76.7 | 36.4 | 59.9 | 73.0 | 78.3 |
| Qwen2-57BA14B-it | 52B | 85.3 | 49.1 | 79.9 | 70.9 | 46.4 | 78.0 | 84.5 | 86.6 |
| **Model Merging and Model Grafting** | | | | | | | | | |
| Task Arithmetic[ICLR23] | 8B | 86.4 | 52.4 | 70.6 | 75.9 | 29.2 | 76.2 | 68.9 | 83.6 |
| Ties-Merging[NeurIPS23] | 8B | 87.2 | 56.3 | 71.1 | 76.1 | 33.9 | 76.7 | 70.1 | 84.0 |
| PCB-Merging[NeurIPS24] | 8B | 87.7 | 56.7 | 71.3 | 76.2 | 34.8 | 76.9 | 70.9 | 84.2 |
| PCB-Merge+DARE[ICML24] | 8B | 88.9 | 57 | 71.5 | 76.2 | 35.0 | 77.2 | 71.2 | 84.6 |
| **Routing based LLM (w/ Self-Specialization)** | | | | | | | | | |
| Self-MoE[ACL25] | 9B | 87 | 49.5 | 70.6 | 71.2 | 38.8 | 77.9 | 67.8 | 84.6 |
| Routed LoRA r512 | 14.1B | 86.4 | 51.4 | 72.5 | 75.9 | 41 | 76.7 | 68.8 | 85.7 |
| Routed LoRA r1024 | 21B | 87.9 | 54.6 | 73.2 | 76.4 | 47.1 | 77.6 | 71.2 | 86.5 |
| TALL-Mask[ICML24] | 16.7B | 90.1 | 58.9 | 72.8 | _76.2_ | 48.6 | 77.8 | 74.6 | 86.4 |
| EMR-Merging[NeurIPS24] | 16.7B | **90.8** | _59.4_ | _73.5_ | 75.7 | 48.9 | 77.9 | **75.8** | _86.9_ |
| Twin-Merging r512[NeurIPS24] | 14.1B | 87.6 | 59.3 | 73.2 | 75.5 | 46.6 | 77.7 | 71.5 | 86.7 |
| Twin-Merging r1024 | 21B | _89.3_ | **60.4** | **73.8** | **76.5** | _49.6_ | **78.4** | _75.3_ | **87.4** |
| **Multi-teacher Distillation** | | | | | | | | | |
| FuseLLM[ICLR24] | 8B | 85.6 | 52.9 | 73.2 | 70.8 | 32.5 | 76.9 | 66.6 | 83.5 |
| FuseChat3.0[ICLR25] | 8B | 88 | 57.2 | 71.3 | 71.8 | **64.2** | **80.2** | 69.4 | 82.2 |
| **Self-Rewarding** | | | | | | | | | |
| Self-Rewarding[ICLR23] | 8B | 84.3 | 49.7 | 68.8 | 74.9 | 46.4 | 77.9 | 66.4 | 83.6 |
| Self-Align[ACL24] | 8B | 85.5 | 47.4 | 69.2 | 73.3 | 47.4 | 78.2 | 65.5 | 83 |
| **Our Approach + Optional Replacement** | | | | | | | | | |
| **SKillWeave (Ours)** | 10B | **91.0**(+0.7) | **62.5**(+3.1) | **75.0**(+1.5) | **77.8**(+1.6) | **52.8**(+3.2) | **79.1**(+0.7) | **76.2**(+0.9) | **88.6**(+1.2) |
| →Self-Rewarding[ICLR23] | 10B | 89.2 | 54.4 | 70.3 | 76.3 | 50.8 | 78.8 | 72.4 | 84.3 |
| →Self-Specialize[ACL24] | 10B | 87.3 | 51 | 70.7 | 72.3 | 35.6 | 76.6 | 68.3 | 85.7 |
| →PEFT | 10B | 86.8 | 49.3 | 73.5 | 74.2 | 47.9 | 78.1 | 68.5 | 86.9 |
| →ASVD | 10B | 89.7 | 60.5 | 73.7 | 76.9 | 47.0 | 78.3 | 74.3 | 87.2 |
| →Delta-Come[NeurIPS24] | 10B | 90.7 | 62.4 | 74.9 | 78 | 52.7 | 79 | 76.3 | 88.4 |

**Model Instantiations.** For most multi-task experiments, we adopt `LLaMA-3.1-8B-Instruct` as our base model, and include results on `Llama-3.2-1B-Instruct` to demonstrate the scalability and efficiency for smaller models. For the LLM-as-an-Agent experiments, we use `Qwen-2.5-7B-Instruct` as the main backbone.

# 5 MAIN RESULTS AND ANALYSIS

## 5.1 GENERAL CAPABILITY RESULTS

Tab. 1 presents the results of general multi-capability evaluation across five categories of baseline methods. Under comparable model sizes, `SkillWeave` consistently achieves the best performance across four out of five task domains, clearly demonstrating its effectiveness for self-improvement.

Notably, a 8B backbone model, enhanced via `SkillWeave` in a fully self-improving manner, outperforms several larger **Open-Source LLMs** by a notable margin (*e.g.,* +8.1 on MATH and +0.6 on GSM8K than Gemma2-27B-it).

**Multi-Teacher Distillation** baselines leverage external teacher models to provide high-quality supervised signals, while not strictly within the self-improvement scope, serve as upper-bound references. Despite this advantage, `SkillWeave` —using only noisy, self-generated data—outperforms these methods in three out of four domains. These improvements can be attributed to *SkillWeave 's core decomposing design*: partitioning the model's capacity into specialized, multi-domain skillpacks. This weaving strategy facilitates efficient use of self-generated data across domains and mitigates cross-task interference, enables more targeted self-improvement.

`SkillWeave` significantly outperforms **Model Merging** and **Model Grafting** methods across all evaluated tasks, due to *our designed merging algorithm*, which integrates diverse skillpacks into a stable and performant shared backbone.

**Routing-based MoE** baselines, such as Twin-Merging and LoRA-MoE, dynamically activate expert modules based on input signals and achieve strong performance. However, `SkillWeave` closes the gap through task-aware model merging and flexible skillpack activation, while incurring significantly lower inference overhead. Moreover, `SkillWeave` surpasses **Self-MoE**—the previous state-of-the-art for self-alignment—by a clear margin of 1.2~10 points.

## 5.2 LLM-AS-AN-AGENT RESULTS

We further evaluate `SkillWeave` in the LLM-as-Agent scenario, with results summarized in Appendix. This setting is particularly well-suited to our modular design: a shared backbone remains active while capability-specific skillpacks are dynamically invoked depending on the user's goal. We compare against two representative deployment strategies:

1. **Task-Specialized LLMs**: Deploying five separate 7B models for five tasks (totaling 5×7B parameters).

2. **A Monolithic Model**: Using a single 32B model capable of handling all tasks jointly.

In contrast, `SkillWeave` requires only a single 7B backbone and five 0.5B skillpacks (totaling 9.5B parameters resident in memory), achieving significant savings in memory footprint. Crucially, the consistent use of a single 7B backbone and S-LoRA implementation for skillpacks allows for effective batch processing. As a result, `SkillWeave` delivers: 1) **4.2× *speedup*** inference than the 32B monolith. 2) **5.5× *speedup*** inference than the 5×7B model deployment. Despite this compact size and significant inference efficiency, `SkillWeave` achieves comparable performance: within 3% of the task-specialized (5×7B) system, and within 5% of the 32B monolithic model.

These results confirm that `SkillWeave` is highly suited for agent-based applications, combining modularity, efficiency, and competitive task performance within a unified framework.

## 6 ADDITIONAL RESULTS

### 6.1 SINGLE-SKILL RESULTS AS ABLATION

In the single-skill setting, we evaluate each skillpack independently on its corresponding task. As shown in Tab. 1, we conduct ablation experiments across two dimensions: the choice of parameter format and the self-improvement algorithm. For each ablation, we replace the corresponding module in our pipeline with a competing alternative, in order to isolate and evaluate the effectiveness of each component in `SkillWeave`.

Our method consistently outperforms **Self-Specialization** across all tasks (*e.g.,*, +11 gain on Math and +7.9 gain on BBH). We also surpass **Self-Rewarding**, a widely adopted prior approach for self-alignment. The key limitation of self-rewarding lies in its reliance on the model's own judgments, which are often inaccurate. These improvements can be attributed to our *selective refine design*: employing rule-based classification to filter synthetic data and leverage Direct Preference Optimization to learn from truly helpful samples while suppress the influence of harmful generations.

Additionally, `SkillWeave` consistently outperform **PEFT**-based methods. These approaches are constrained by their limited trainable parameter space, suggesting that *full-parameter adaptation remains essential* for achieving strong task specialization.

Compared to **ASVD**, our approach are observed a 30% advantage in all domains. This further validates *our full-tuning-then-zip design* of performing full-parameter fine-tuning followed by aggressive quantization, rather than relying solely on delta-based adapters. And, compared to **DeltaCome**, a recent delta-compression method, our approach achieves superior performance on 5 out of 8 tasks.

## 6.2 SKILLZIP RESULTS

**Settings.** We evaluate our quantization strategy under various configurations, each denoted as $X_{k_1}A_{k_2}B_{k_3}$. For instance, $X_8A_4B_4$ denotes using 8-bit activation $X$, 4-bit $A$, and 4-bit $B$. The SKillZip configurations include $X_8A_8B_8$, $X_8A_4B_4$, $X_4A_4B_8$, and $X_4A_4B_4$. We compare our method against prior compression baselines: BitDelta (denoted as $X_{16}W_1$), ASVD($X_{16}A_{16}B_{16}$), and DeltaCome. All experiments are conducted using Llama3.1-8B as the base model, with three model separately finetuned on three alignment tasks. Full results are presented in Tab. 2 and Figure 3.

**Ablation Study.** We perform an in-depth ablation analysis on four technical components of our SKillZip framework: *1)* Model *merg*ing for unified knowledge aggregation, *2)* Matrix *concat*enation, *3)* Channel-wise *smooth*ing, *4)* Rank-wise *rota*tion. We test combinations of these components and evaluate their contributions individually. Results in Tab. 2 show that each component contributes to performance gains, with the full combination yielding an average +2.1 improvement over the base quantization.

Table 2: Ablation study on SKillZip and its comparison with other delta-compression baselines. We evaluate compression fidelity across multiple settings and methods.

| Setting | Merge | Conca. | Smooth | Rotate | MATH | MBPP | GPQA | Average↑ |
|---|---|---|---|---|---|---|---|---|
| Base | - | - | - | - | 51.9 | 66.9 | 33.6 | 50.8 |
| Target | - | - | - | - | 67.8 | 73.9 | 38.4 | 60.0 |
| $X_8A_8B_8$ | ✓ | ✓ | ✓ | ✓ | **67.6** | **73.8** | **38.3** | **59.8** |
| $X_8A_8B_8$ | ✓ | ✓ | ✓ | ✗ | 66.6 | 73.6 | 38.0 | 59.4 |
| $X_8A_8B_8$ | ✓ | ✓ | ✗ | ✗ | 63.0 | 72.9 | 37.2 | 57.7 |
| $X_8A_8B_8$ | ✗ | ✓ | ✓ | ✓ | 67.1 | 73.6 | 38.1 | 59.6 |
| $X_8A_8B_8$ | ✓ | ✗ | ✓ | ✓ | 67.0 | 73.6 | 38.2 | 59.6 |
| $X_8A_4B_4$ | ✓ | ✓ | ✓ | ✓ | 67.1 | 73.5 | 38.0 | 59.5 |
| $X_4A_4B_8$ | ✓ | ✓ | ✓ | ✓ | 66.1 | 73.2 | 37.8 | 59.0 |
| $X_4A_4B_4$ | ✓ | ✓ | ✓ | ✓ | 66.0 | 73.2 | 37.7 | 58.9 |
| DeltaCome[NeurIPS24] | | | | | 67.2 | 73.6 | 38.3 | 59.7 |
| $X_{16}A_{16}B_{16}$(SVD) | | | | | 61.0 | 71.1 | 36.2 | 56.3 |
| $X_{16}A_8B_8$(from DeltaCome) | | | | | 63.0 | 72.1 | 37.0 | 57.4 |
| $X_{16}A_4B_4$(from DeltaCome) | | | | | 65.4 | 73.2 | 37.6 | 58.7 |
| $X_{16}W_1$(BitDelta[NeurIPS23]) | | | | | 63.2 | 72.8 | 37.3 | 57.7 |
| $X_{16}A_{16}B_{16}$(ASVD) | | | | | 623 | 72.6 | 37.3 | 57.0 |

**Fidelity and Latency.** As summarized in Tab. 2, our quantized skill-packs maintain comparable or better performance than previous approaches. Specifically, we achieve:

- A 2% performance gain over Delta-Come,

- A 12% improvement over BitDelta.

- A 15% improvement over ASVD.

More importantly, our approach offers significant inference efficiency advantages. Figure 3 illustrates kernel-level latency (right) and end-to-end latency (left) across all methods. Thanks to our *full quantization* of both weights and activations, we are able to directly compute in INT8 or INT4 formats using tensor core acceleration. This yields:

- 1.38× speedup over DeltaCome,

- 1.04× speedup over S-LoRA.

Although BitDelta uses even lower-bit quantization, its runtime dequantization and rehydration steps result in slower inference. Our approach slightly underperforms BitDelta in

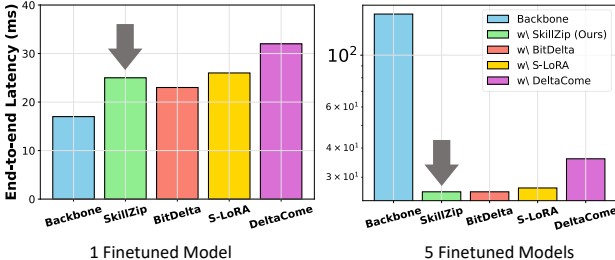

Figure 3: End-to-end inference latency comparison of delta-compression methods on a single A100-80G GPU using Qwen2.5-7B and its finetuned variants. The right plot shows the average token generation time for a single finetuned model, while the left plot reports the latency when serving five finetuned models concurrently. End-to-end latency includes full decoding time per token. And "backbone" denotes directly deploying multiple finetuned models simultaneously.

raw kernel latency (by a marginal 0.08×), but vastly outperforms it in overall task accuracy. This represents a superior trade-off between fidelity and latency, making our method both deployment-friendly and high-performing.

# 7 CONCLUSION

We present `SkillWeave`, a modular framework for self-improvement of LLMs without external supervision. By decomposing task capabilities into skill-specific modules and refining them via preference-based optimization, our approach enables scalable and interpretable self-improvement. Further, we introduce `SKillZip`, a fully quantized delta-compression method that allows efficient deployment of specialized skills. Experiments across diverse benchmarks demonstrate that `SkillWeave` achieves superior performance and efficiency compared to existing self-training and model merging baselines. Our work offers a practical path toward continual and modular enhancement of LLMs using only their own outputs.

## ETHICS STATEMENT

This research adheres to established ethical standards in artificial intelligence and machine learning. All experiments were conducted using publicly available datasets or models under their respective licenses, and no personally identifiable or sensitive information was involved. The methods proposed are intended for academic and scientific purposes, with the goal of advancing understanding in machine learning rather than deployment in high-stakes decision-making without further safeguards. We recognize that advances in AI systems may pose potential societal risks, including issues of fairness, misuse, privacy, and environmental impact due to computational resource consumption. To mitigate these concerns, we emphasize responsible reporting of results, transparent acknowledgment of limitations, and a clear separation between research contributions and downstream applications. Future work building on this research should continue to assess possible ethical implications, particularly regarding bias, safety, and dual-use risks, and adopt appropriate measures to ensure beneficial and equitable outcomes.

## REPRODUCIBILITY STATEMENT

Our implementation, including all code, training scripts, and evaluation datasets, is available at: https://anonymous.4open.science/r/anonymous-repo-B220

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

APPENDIX

# Appendix for Skill Weaving

## OVERVIEW

This paper proposes a efficient self-improvement approach that decompose general-purpose LLMs into a collection of *SkillPacks* that reorganize and refine the model's internal capabilities. The appendix is structured according to our key contributions. We also make the project code available via an anonymous link for reproducibility: anonymous-repo

- Appendix B (Implementation Details) provides a detailed implementation description of rule-based verification and SKillZip design.
- Appendix C (Additional Results) includes the results of agentic evaluation and results on smaller LLM.
- Appendix D (Related Works and Baselines) provides a detailed description of baseline and related works.
- Appendix E (Evaluation details) outlines the evaluation benchmarks and training datasets.
- Appendix F (Limitation and Future Work) provides a detailed dataset description.

## A  THE USE OF LARGE LANGUAGE MODELS (LLMs)

Throughout the preparation of this manuscript, large language models were employed exclusively for light stylistic refinement and the occasional grammatical adjustment. Every conceptual insight analytical thread, and interpretive conclusion emerged from the authors themselves; no algorithmic assistance was solicited for the framing, design, or substance of the work, and full scientific responsibility rests with the human contributors alone

## B  IMPLEMENTATION DETAILS

### B.1  RULE-BASED VERIFICATION

To ensure the quality of self-generated data used for preference optimization, we design task-specific rule-based verification strategies tailored to each domain and dataset.

**Mathematics.**  For math-related datasets (e.g., GSM8K, MATH), we extract the final numerical answer from model outputs using regex-based pattern matching and compare it against the ground-truth solution. Only completions with exact matches are considered "helpful," while mismatches are marked as "harmful."

**Code.**  In code generation tasks (e.g., HumanEval, MBPP), we execute the model-generated programs in a secure sandbox environment. A sample is accepted only if it passes all test cases and its output matches that of the reference solution. We also detect exceptions or infinite loops to identify invalid generations.

**Reasoning.**  For tasks involving open-ended reasoning (e.g., ARC, BBH), we combine answer correctness with lightweight heuristic filters. These include rejecting overly short or excessively verbose answers, and favoring logically structured completions with sufficient diversity and specificity.

**Dialogue.**  For instruction-following dialogue tasks (e.g., IFEval, AlpacaEval), we adopt two verification protocols for each type of prompts:

1) We design diverse prompting formats with explicit instruction constraints (e.g., "mention the keyword 'AI' at least 3 times"), and verify completions against corresponding structural rules—covering aspects such as keyword frequency, maximum length, repetition, banned tokens, format, paragraph

structure, language tone and so on. 2) For loosely defined or open-ended instructions, we employ ensemble scoring from two widely adopted reward models to assess the quality of model completions. Each completion is evaluated across overall quality, and helpfulness, and harmlessness. We discard outliers whose average scores fall below the 10th percentile or exceed the 90th percentile to ensure balance and avoid preference bias. The reward models used are: 1) RewardModel-Mistral-7B-for-DPA-v1 [1] 2) RLHFlow/ArmoRM-Llama3-8B-v0.1 [2]

**Agent Tasks.** For structured agent benchmarks such as **AgentBench** and **LifelongAgentBench**, we inherit and reuse the official rule-based evaluation criteria of each subdomain. For example, in the **Database** domain, a generation is marked correct only if the execution result of the generated SQL query matches the gold-standard output. Similar logic applies to **Operating System**, **Web**, and **Knowledge Graph** domains.

These rule-based verifications are fully automated and domain-specific, enabling efficient filtering of low-quality synthetic data prior to DPO training.

### B.2 SKILLZIP IMPLEMENTATION

**Hardware-Compatible Scaling Constraints.** In hardware-accelerated GEMM (Lawson et al., 1979) kernels, FP16 scaling can only be efficiently applied along the outer dimensions—that is, before and after the INT8 matrix multiplication. In our setting, this means the quantized multiplication: $\text{diag}(\vec{s}_A) \cdot \hat{A} \cdot \hat{B} \cdot \text{diag}(\vec{s}_B)$, where $\hat{A}$ and $\hat{B}$ are INT8 or INT4 matrices, and $\vec{s}_A, \vec{s}_B$ are FP16 scaling vectors applied at the input and output channels, respectively. Crucially, scaling between the two matrix multiplications (i.e., $\hat{A} \cdot \text{diag}(\vec{s}_A) \cdot \text{diag}(\vec{s}_B) \cdot \hat{B}$) is not supported by Tensor Core hardware and leads speed degradation. Hence, we constrain quantization granularity as follows: per-token or per-tensor quantization for $\hat{X}$, per-tensor for $\hat{A}$, and per-tensor or per-channel for $\hat{B}$. The reconstructed output is computed by:

$$Y = s_A \cdot \text{diag}(\vec{s}_X) \cdot \hat{X}\hat{A}\hat{B} \cdot \text{diag}(\vec{s}_B). \tag{5}$$

where $s_A$ is a per-tensor scalar scaling factor and can be merged with $\vec{s}_X$ as $\text{diag}(s_A\vec{s}_X)$.

**Outlier-Induced Quantization Error.** In the field of quantization, accuracy degradation often arises due to outliers — a small number of activation channels whose magnitudes are 100× larger than the rest. When misaligned with quantization axes, these outliers inflate the dynamic range, forcing most values in each quantization block to collapse to zero, thereby leading to significant quantization error.

Consider the activation matrix $X \in \mathbb{R}^{T \times C_i}$, where each row corresponds to a token and each column corresponds to an input channel. In many LLMs activation, certain channels (say, the $i$-th column $X_{:,i}$) contain values with magnitudes exceeding 2000 across most or all tokens—this forms a channel-wise outlier. However, due to hardware constraints, activation quantization must be applied *per-token*—i.e., along the row dimension, orthogonal to the direction in which outliers occur. This misalignment creates a quantization dilemma: each row vector $X_{j,:}$ acting as an independent quantization block, spans both extremely large outlier dimensions and low-magnitude, typical values (e.g., in the range $[-15, +15]$). When such a mixed-range vector is quantized uniformly into 8 bits, the outlier values dominate the scale and are mapped to the maximum bin (e.g., ±127), leaving the remaining values compressed into a narrow range near zero. As a result, most non-outlier values collapse to zero after quantization, effectively erasing useful information and causing severe precision loss.

A similar problem arises when quantizing low-rank weights $W \approx AB$ Singular value decomposition (SVD) often concentrates most of the energy into the top few singular directions (i.e., the leading columns of $A$ and corresponding rows of $B$) forming rank-wise outliers, while the quantization is applied independently across matrix rows or columns.

**Double Smoothing Strategy.** To mitigate this, we propose a double smoothing strategy.

---

[1] https://huggingface.co/RLHFlow/RewardModel-Mistral-7B-for-DPA-v1
[2] https://huggingface.co/RLHFlow/ArmoRM-Llama3-8B-v0.1

*(1). Channel-wise Smoothing.* We compute a domain-specific channel scaling vector $\vec{s} \in \mathbb{R}^{C_i}$ based on the average absolute activation magnitude per channel: $\vec{s}_i = f(\text{mean}(|X_{:,i}|))$, where $f(\cdot)$ is a monotonic mapping tuned for aggressive smoothing. We rescale: $X \leftarrow X \cdot \text{diag}(s^{-1}), W \leftarrow \text{diag}(s) \cdot W$ thus transferring channel-wise quantization difficulty from $X$ to $W$. Unlike SmoothQuant (Xiao et al., 2023), we apply more aggressive compression of outlier dimensions because: (1) the quantization burden is now shared across $X, A, B$ and each only bear one-third of the burden; and (2) each task domain $\tau_i$ has distinct activation statistics, for which we fit a unique scaling vector $\vec{s}_i = f(X^i)$

*2. Rank-wise Smoothing.* we apply truncated singular value decomposition (SVD) to the smoothed weight matrix $W_{\text{smooth}}$: $W_{\text{smooth}} \approx U_R \Sigma_R V_R^T$, $A = U_R \Sigma_R^{1/2}$, $B = \Sigma_R^{1/2} V_R^T$. And then apply a second-stage rank-wise smoothing: a appropriate orthogonal rotation matrix $Q$ to disperses the energy evenly across dimensions: $A_{\text{rot}} = AQ$, $B_{\text{rot}} = Q^T B$. The optimal $Q$ should satisfy:

- $AQ$ and $Q^T B$ are individually easy to quantize;
- $X_{\text{smooth}} AQ$ is uniform, avoiding inner-product alignment between dominant rows of $X_{\text{smooth}}$ and columns of $AQ$, reducing precision loss when INT32 is truncated to INT8.

In practice, we find that randomly sampled orthogonal matrices suffice. We sample 10 candidates and select the one minimizing the final quantization loss.

Similar to DeltaCome (Ping et al., 2024), we also adopt GPTQ quantization (Frantar et al., 2022) in the final step.

**Hardware-Aware Inference.** At inference time, we follow a fully quantized pipeline optimized for Tensor Core execution:

1) Input $X$ is first smoothed using precomputed $s^{-1}$, and quantized to INT8/INT4 $\hat{X}_{\text{smooth}}$.

2) We then load the quantized matrix , and perform the first INT8 GEMM: $\hat{X}_{\text{smooth}} \cdot \hat{A}_{\text{smooth}} \rightarrow$ INT32. The output is intermediate result in INT32 format.

3) Rather than dequantizing the INT32 result, we aggressively truncate it to INT8—thereby preserving throughput and maintaining compatibility with the next GEMM. This truncated INT8 matrix is then used as input to a second matrix multiplication with $\hat{B}_{\text{smooth}}$, again using Tensor Core acceleration: INT8 $\cdot \hat{B}_{\text{smooth}} \rightarrow$ INT32. While, this second multiplication incorporates scaling vectors to recover the final FP16 output scale.

4) All scaling vectors and quantization parameters are precomputed to minimize runtime overhead.

5) To avoid latency from separate dequantization kernels, we fuse the dequantization step into the GEMM computation (Wang et al., 2024b), utilizing fast-dequantization strategies (Kim et al., 2022).

This two-stage smoothing and hardware-aware execution pipeline forms the core of SkillZip, enabling low-latency, high-accuracy inference across diverse domains using low-rank, low-bit skillpacks.

## B.3 ROUTING IMPLEMENTATION

This section provides additional implementation details on how SkillWeave performs domain routing during inference, how routing models are trained, and how routing accuracy affects downstream performance. We additionally quantify the routing overhead on inference speed. These clarifications address concerns regarding domain identification when activating skillpacks at inference time.

SkillWeave adopts a routing mechanism fundamentally different from Mixture-of-Experts (MoE) architectures: the backbone network is always activated, while a skillpack is activated *only* if the input belongs to its corresponding domain. As a result, SkillWeave does not perform token-level expert selection but instead uses domain-level routing.

**Agent Tasks.** For agent-style tasks (e.g., tool selection, structured multi-step workflows), the domain type of each request is inherently known in advance. Tool invocation APIs explicitly specify which capability is being queried, and each skillpack corresponds to one such tool-specific or capability-specific domain. Therefore, no domain classification is required, and routing is deterministic.

**General Tasks.** For the general tasks evaluated in Table 1, we consider two settings:

1. **Oracle Domain Labels.** For fair comparison with prior work, the main table assumes known domain labels, which is standard for multi-domain evaluation.

2. **Learned Router.** To approximate realistic scenarios where domain labels may be unknown, we additionally train a domain classifier that predicts the skillpack to activate at inference time.

**Training the Routing Model** We adopt Qwen2.5-0.5B as a lightweight routing model and attach a linear classification head. The model is fine-tuned for sequence classification using 60,000 labeled prompts across domains. Training is performed for 3 epochs on L40S GPUs, taking approximately one hour.

Routing accuracy is summarized in Table 3. The router achieves exceptionally high performance across all domains, with accuracy, precision, and recall all exceeding 0.999. Both false positive rate (FPR) and false negative rate (FNR) remain below 0.001. These results indicate that domain misclassification is exceedingly rare.

Table 3: Routing model performance across domains.

| Domain | Accuracy | FPR | FNR | Precision | Recall | F1 score |
|---|---|---|---|---|---|---|
| Mathematics | 0.9987 | 0.0001 | 0.0013 | 0.9993 | 0.9987 | 0.9990 |
| Coding | 1.0000 | 0.0000 | 0.0000 | 1.0000 | 1.0000 | 1.0000 |
| Dialogue | 0.9990 | 0.0002 | 0.0010 | 0.9990 | 0.9990 | 0.9990 |
| Reasoning | 1.0000 | 0.0001 | 0.0000 | 0.9992 | 1.0000 | 0.9996 |

**Impact of Routing on Downstream Task Performance** We compare the downstream performance under two routing conditions:

- **Oracle routing**: using ground-truth domain labels.
- **Model routing**: using the learned routing model.

Table 4 shows that the performance difference between the two settings is negligible across all benchmarks. This confirms that routing accuracy is sufficiently high that it does not affect final task performance.

Table 4: Downstream performance with oracle vs. learned routing.

| Method | Mathematics | | Coding | | Dialogue | | Reasoning | |
|---|---|---|---|---|---|---|---|---|
| | GSM8k | MATH | HumanEval | MBPP | AlpacaEval2 | IFEval | BBH | ARC-C |
| **SKillWeave (Oracle Routing)** | 91.0 | 62.5 | 75.0 | 77.8 | 52.8 | 79.1 | 76.2 | 88.6 |
| **SKillWeave (Learned Routing)** | 91.0 | 62.3 | 75.0 | 77.8 | 52.9 | 78.9 | 76.2 | 88.6 |

**Case Study.** We manually examined all samples where the router's prediction differs from the oracle domain. Interestingly, misclassified samples often contain mixed-domain content, such as a dialogue prompt involving embedded mathematics. In such cases, the mathematical skillpack can solve the underlying subproblem better than the dialogue model. Figure 4 illustrates an example where routing to the "Math" skillpack yields higher-quality output than routing to the "Dialogue" skillpack. This explains why routing mistakes do not degrade overall accuracy.

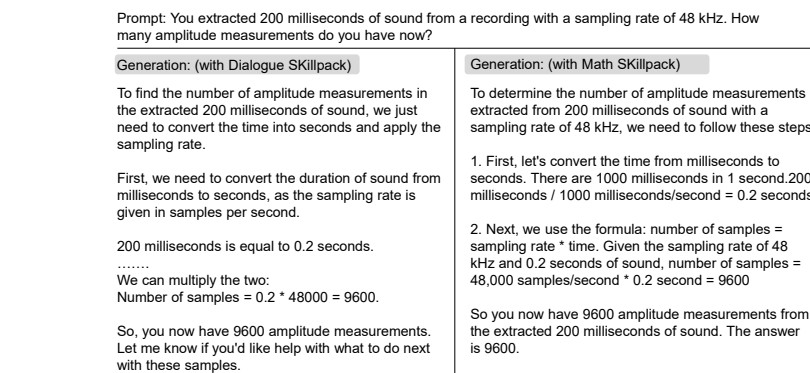

Figure 4: An example of misidentification of learned router.

**Routing Overhead During Inference**  We further measure the inference-time overhead of routing. As reported in Appendix E.6, routing increases latency only marginally. The slowdown stems primarily from **GPU memory contention** due to the routing model being resident in memory, rather than from the routing model's prefill or forward-pass computation.

Importantly, the routing model runs only once per request—not per token—so its cost is effectively negligible compared to backbone decoding.

## C  ADDITIONAL RESULTS

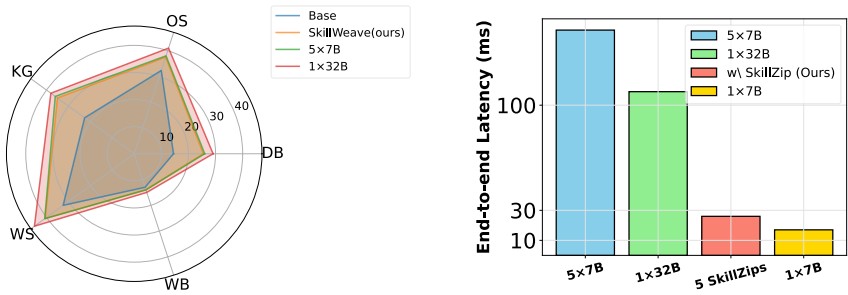

Figure 5: The left panel reports the performance across five dimensions of AgentBench under different configurations. The right panel compares the end-to-end inference latency when simulating agentic service calls, measured on a single A100-80G GPU. The 5×7B setting denotes five independently fine-tuned Qwen2.5-7B-Instruct models, each specialized for a different agent task, while 1×32B refers to a single monolithic Qwen2.5-32B-Instruct model.

### C.1  LLM-AS-AGENT RESULTS

Figure 5 illustrates the evaluation of SkillWeave in the LLM-as-Agent setting using Qwen2.5-7B-Instruct. The 5×7B configuration refers to five independently fine-tuned 7B models, each specialized for a distinct agent task, while 1×32B denotes a single monolithic Qwen2.5-32B-Instruct model. SkillWeave not only achieves significant inference acceleration, but also maintains competitive task performance, as detailed in the main text.

### C.2  RESULTS ON OTHER MODEL

Table 5 reports the general multi-capability evaluation results on LLAMA3.2-1B-INSTRUCT, comparing SkillWeave against multiple baseline approaches. Under comparable model sizes, SkillWeave consistently outperforms all four baselines, clearly demonstrating its robustness and effectiveness in low-resource settings.

Table 5: Overall performance of SkillWeave in the general capability setting using Llama-3.2-1B-Instruct as the backbone. The best results for baselines are shown in bold and the second best are underlined.

| Method | #Params | Mathematics | | Coding | | Dialogue | | Reasoning | |
|---|---|---|---|---|---|---|---|---|---|
| | | GSM8k | MATH | HumanEval | MBPP | AlpacaEval2 | IFEval | BBH | ARC-C |
| Llama3.2-1B-Instruct | 1.15B | 45.8 | 32.3 | 38.9 | 49 | 9.7 | 58.4 | 32.4 | 69.4 |
| self-rewarding | 1.15B | 54.2 | 32.6 | 38 | 44.6 | 11.9 | 63.2 | 30.4 | 71.1 |
| Twin-merging | 1.48B | 53.4 | 33.8 | 39.6 | 49.4 | 11.5 | 61.9 | 35.2 | 70.1 |
| FuseChat3.0 | 1.48B | **57.4** | **36.2** | 41.4 | 45.5 | **27.1** | **72.5** | **45.8** | **77.5** |
| **SKillWeave(Ours)** | 1.42B | 56.7 | 34.9 | **42.5** | **49.7** | 12.5 | 64.1 | 36.4 | 72.9 |
| →PEFT | 1.42B | 46.1 | 32.0 | 39.1 | 48.2 | 12.1 | 62.7 | 34.8 | 70.8 |

The self-rewarding approach performs poorly due to the limited evaluation ability of small models acting as judges, highlighting the necessity of our *rule-based verification* framework for reliable self-improvement. Meanwhile, PEFT-based methods lag far behind—even on small models—due to their constrained trainable parameter space, further validating the superiority of our *fully-finetune-then-compress (SkillZip)* strategy.

Although SkillWeave underperforms FuseChat3.0 in many domains, this is largely attributable to FuseChat3.0 leveraging stronger teacher models, which significantly outperform the LLAMA3.2-1B-INSTRUCT student.

## C.3 INFERENCE LATENCY MEASUREMENT AND ADDITIONAL RESULTS

This section provides detailed explanations of our end-to-end inference latency measurement protocol. We additionally include standardized latency tables reporting throughput (tokens/s), normalized latency (ms/token), and kernel-level measurements across different batch sizes and sequence lengths.

### C.3.1 EXPERIMENTAL SETUP

**Settings.** All latency experiments are conducted on a single *NVIDIA A100-80G* GPU. As to inference engine, We adopt *S-LoRA*Sheng et al. (2023) as our primary backend, because its implementation is most convenient for integrating our optimized low-rank kernels. Our methodology is compatible with vLLM and sGLang, which support similar execution models. The base model is *Llama3.1-8B-Instruct*. The testing Skillpack matrices have an average effective *rank of 400* (ranging from 300 to 600 depending on the module). As to request arrival process, we follow prior latency studies. The request stream is generated using a *Gamma arrival process* with coefficient of variation = 1, which produces highly bursty arrival patterns representative of real-world workloads. All measurements in Figure 3 and Figure 4 use this process.

**Latency metrics.** For each configuration we compute:

- **Request throughput** (req/s)
- **Token throughput** (tokens/s)
- **Latency per generated token** (ms/token)
- **Time-to-First-Token** (TTFT)
- Distribution statistics: mean, median, P95, P99, min, max

### C.3.2 BASELINES DETAILS

For the "Backbone" setting in Figure 3 and the "5×7B" baseline in Figure 4, we deploy multiple finetuned models using *vLLM*, each as an independent worker process under NVIDIA MPS. GPU memory is *statically partitioned* among workers according to their average request rate.

Because a single A100-80G cannot host more than five 7B–8B models simultaneously, we follow a realistic eviction-and-reloading policy: (1) track domain frequencies for past and projected future requests; (2) evict the model with the lowest combined usage score; (3) load the model with the highest expected usage into the freed memory.

We include this strategy for fairness: any alternative (e.g., sequential swapping) would only increase latency. Due to severe queuing delays and long tail delays, this strategy has already meant complete failure in reality, further highlighting the advantage of SkillWeave.

### C.3.3 END-TO-END LATENCY UNDER DIFFERENT PROMPT AND GENERATION LENGTHS

We evaluate three groups of scenarios (full results in Table 6): (1) Varying prompt length. (2) Varying generation length. (3) Varying request rate.

Table 6: End-to-end latency comparison.

| Settings | | | 5×8B | | | with 5 Skillpacks | | | | | | | |
|---|---|---|---|---|---|---|---|---|---|---|---|---|---|
| request rate | prompt length | generation length | request throughput | token throughput | Latency (mean) | request throughput | token throughput | Latency (mean) | Latency (median) | Latency (P95) | Latency (P99) | Latency (min) | Latency (max) |
| 5 | 50 | 50 | 1.84 | 101.98 | 2.02 | 4.94 | 274.18 | 0.75 | 0.75 | 0.76 | 0.76 | 0.69 | 0.77 |
| 5 | 200 | 50 | 1.43 | 91.19 | 2.62 | 4.94 | 314.60 | 0.76 | 0.75 | 0.76 | 1.09 | 0.69 | 1.10 |
| 5 | 500 | 50 | 1.39 | 90.09 | 2.70 | 4.94 | 321.12 | 0.76 | 0.76 | 0.76 | 0.77 | 0.69 | 0.77 |
| 5 | 100 | 20 | 2.08 | 51.02 | 0.75 | 4.97 | 121.86 | 0.31 | 0.31 | 0.32 | 0.32 | 0.30 | 0.32 |
| 5 | 100 | 100 | 1.43 | 159.23 | 5.15 | 4.89 | 543.92 | 1.51 | 1.51 | 1.53 | 1.53 | 1.35 | 1.53 |
| 5 | 100 | 200 | 1.28 | 307.48 | 11.44 | 4.78 | 1149.18 | 3.06 | 3.07 | 3.09 | 3.09 | 2.72 | 3.10 |
| 5 | 500 | 1000 | 0.46 | 597.20 | 103.84 | 2.32 | 2983.21 | 20.79 | 21.33 | 22.33 | 22.40 | 16.53 | 22.41 |
| 5 | 500 | 4000 | 0.23 | 920.68 | 1857.59 | 1.42 | 5691.31 | 300.50 | 300.49 | 300.95 | 301.00 | 300.00 | 301.09 |
| 0.5 | 500 | 4000 | 0.03 | 122.13 | 1411.42 | 0.14 | 572.80 | 300.94 | 300.94 | 300.99 | 301.09 | 300.88 | 301.09 |
| 1 | 500 | 4000 | 0.06 | 238.64 | 1441.09 | 0.29 | 1142.88 | 300.91 | 300.91 | 300.99 | 300.99 | 300.83 | 301.09 |
| 3 | 500 | 4000 | 0.15 | 612.56 | 1677.45 | 0.85 | 3419.07 | 300.53 | 300.55 | 300.97 | 300.99 | 300.00 | 301.09 |
| 5 | 500 | 4000 | 0.23 | 920.68 | 1857.59 | 1.42 | 5691.31 | 300.50 | 300.49 | 300.95 | 301.00 | 300.00 | 301.09 |
| 7 | 500 | 4000 | 0.29 | 1162.21 | 2046.38 | 1.98 | 7914.48 | 300.50 | 300.50 | 300.95 | 300.99 | 300.00 | 301.09 |

These experiments quantify end-to-end throughput and latency of SkillWeave and naive baseline across a wide range of realistic workloads.

### C.3.4 KERNEL-LEVEL LATENCY: PREFILL AND DECODE

We further measure the **per-kernel latency** of our optimized low-rank compute path. Using the Llama3.1-8B-Instruct `down_proj` matrix of shape $4096 \times 14336$, with rank 600, we benchmark:

- the **prefill** phase with each sequence length $= 1000$
- the **decode** phase with each sequence length $= 1$
- batch sizes $\{1, 4, 10, 32\}$

Table 7 compares the standard **FP16 (X16W16)** matrix multiplication kernel against our optimized **X8A8A8** kernel with static quantization/dequantization. We also report performance under CUDA Graphs.

Table 7: Kernel latency comparison (ms). Left: FP16 X16W16. Right: Optimized X8A8A8.

| Batch | X16W16 | | | X8A8A8 | | |
|---|---|---|---|---|---|---|
| | **Prefill** | **Decode** | **Decode (Graph)** | **Prefill** | **Decode** | **Decode (Graph)** |
| 1 | 0.6673 | 0.1167 | 0.1086 | 0.1698 | 0.0558 | 0.0271 |
| 4 | 2.0216 | 0.1064 | 0.0945 | 0.4669 | 0.0525 | 0.0290 |
| 10 | 4.9063 | 0.1064 | 0.0991 | 1.1168 | 0.0524 | 0.0291 |
| 32 | – | 0.1082 | 0.0963 | – | 0.0553 | 0.0280 |

The optimized kernel is significantly faster than the backbone's FP16 path, and the incremental latency introduced by activating a single skillpack is negligible.

## D   RELATED WORKS AND BASELINES

This section provides detailed descriptions of the related works and baseline methods used in our study. Some of these works also serve as baselines in our experiments and are marked with a superscript ASTERISK (*) next to their names.

### D.1   SELF-IMPROVEMENT FOR LLM

Self-improvement methods aim to enhance language models using only their own generations, without relying on external human annotations or teacher supervision. These approaches explore how LLMs can autonomously refine their capabilities via self-generated data and internal feedback mechanisms.

- **Self-Specialization*** (Kang et al., 2024b) finetunes an LLM on its own task-specific generations to induce latent expertise. However, it does not distinguish between high- and low-quality outputs, resulting in unstable performance and potential error reinforcement.
- **Self-rewarding*** (Yuan et al., 2024) introduces LLM-as-a-judge prompting, where the model scores its own responses using handcrafted criteria and then applies preference optimization (e.g., DPO) to favor high-quality generations.
- **Meta-rewarding*** (Wu et al., 2024) further enhances the self-rewarding method by improve both acting and judging skills of models simultaneously.
- **Self-Align*** (Sun et al., 2023) presents human-written alignment principles to the model, which are then internally triggered to filter and guide generation. It offers a lightweight and interpretable way to regulate self-improvement using task-level rules.
- **Self-MoE*** (Kang et al., 2024a) extends Self-Specialization into a modular multi-task setting by training separate LoRA experts for each domain. These LoRA modules are then assembled into a LoRA-MoE model (in the LoRAHub style) to achieve compositional generalization across tasks.
- **RLAIF** (Lee et al., 2024) reuses the RLHF pipeline but replaces reward signals with synthetic preferences generated by LLMs themselves. It provides a scalable way to apply reinforcement learning without manual annotations.

We adopt **Self-Rewarding** and **Self-Align** as multi-skill baselines in our experiments. Following their original setups, we design distinct prompting templates and scoring criteria for each task domain. However, instead of training each domain in isolation, we merge all task datasets and perform joint training using both SFT and DPO objectives over the combined data—ensuring consistency and comparability with SkillWeave.

We also include **Self-Rewarding** and **Self-Specialization** as single-skill baselines, replacing the self-improvement module in our pipeline with these alternatives. For fairness, we retain our full fine-tuning and SkillZip compression stages, allowing us to isolate and evaluate the effectiveness of the self-improvement component itself. The corresponding results are reported in Table.1.

### D.2   MODEL MERGING AND GRAFTING

Model Merging aims to combine multiple task-specific models into a single multitask model by algebraically manipulating their finetuned parameters. By scaling and summing the deltas from different tasks, merging methods seek to amplify beneficial knowledge while mitigating harmful interference across tasks. These approaches attempt to resolve parameter conflicts and redundancies to form a unified and robust multitask representation. Model Grafting (Panigrahi et al., 2023), on the other hand, takes a more surgical approach. It selectively transplants a small subset of task-specific parameters into the pre-trained model, aiming to recover finetuned performance while introducing minimal overhead. This paradigm emphasizes the localization and reuse of transferable skills across tasks.

- **Task Arithmetic*** (Ilharco et al., 2023) first introduces the concept of *"task vectors"*—the difference between a finetuned model and its base—and proposes to merge them via linear operations: $\theta_{\text{merge}} = \theta_{\text{init}} + \lambda * \sum_{t=1}^{n} \tau_t$, where $\tau_t$ is the task vector for task $t$.

- **AdaMerging** (Yang et al., 2024a) extends task arithmetic by automatically learning optimal linear merging coefficients to adaptively tune layer-wise or task-wise weights. It uses entropy minimization on unlabeled evaluation data as a surrogate objective, enabling unsupervised merging.

- **Ties-Merging\*** (Yadav et al., 2024) further solves the task conflict problem in Task Arithmetic (Ilharco et al., 2023) by explicitly resolving parameter conflicts via a three-stage process: Trim redundant parameters, Elect, and Disjoint Merge to isolate interference.

- **PCB-Merging\*** (Du et al., 2024) effectively adjusts parameter coefficients through balancing parameter competition within model population.

- **FR-Merging** (Zheng & Wang, 2024) emphasizes the importance of merging common capabilities into the backbone before combining task-specific skills, thereby preserving general knowledge.

- **DARE\*** (Yu et al., 2023) sets the majority of delta parameters to zero and rescale the rest by $\theta' = \theta \cdot (1/(1-p))$ where $p$ is the proportion of delta parameters dropped, therefore efficiently reduces parameter redundancy.

- **TALL-MASK\*** (Wang et al., 2024a) localize the task-specific information in a multi-task vector, which deactivates irrelevant parts for each task in the merged multi-task vector with binary masks.

- **EMR-Merging\*** (Huang et al., 2024) first selects a unified model from all weights, then generates lightweight task-specific modulators—masks and rescalers—to align direction and magnitude with each source model.

- **Model Grafting** (Panigrahi et al., 2023) introduces the notion of skill localization by identifying which parameter subsets are critical for each task. It then selectively grafts these modules onto the pre-trained model, achieving task recovery without full model duplication.

In our evaluation, all model merging and grafting baselines are instantiated using the same pre-trained model and identical task-specific finetuned models produced by SkillWeaving. To ensure fair comparison, we report that TALL-MASK and EMR-Merging introduce significantly more parameters than SkillWeave due to the inclusion of large task-specific components.

## D.3    LoRA-based MoE

LoRA-based Mixture-of-Experts (MoE) models combine the modularity and specialization advantages of MoE architectures with the parameter efficiency of Low-Rank Adaptation (LoRA). In these models, each expert is implemented as a lightweight LoRA adapter, typically applied to all major modules within every Transformer block. This design enables fine-grained task specialization while significantly reducing memory and training overhead compared to traditional dense experts.

- **LoRA-MoE\*** (Gao et al., 2024) extends the standard MoE framework by deploying multiple LoRA experts in parallel across a multi-task training setup. It introduces an expert-balancing mechanism to ensure that all LoRA modules are utilized effectively.

- **LoRAHub** (Gao et al., 2024) employs Low-rank Adaptations to dynamically combine task-specific modules for cross-task generalization, and adapts to new tasks by configuring $\theta' = \sum_{k=1}^{K} w_k \cdot \theta_k$.

- **Twin-Merging\*** (Lu et al., 2024) compress multiple finetuned models into a compact LoRA-MoE format by merging, singular value decomposition and pruning. Then a trainable router is used to dynamically select among them.

## D.4    Delta Compression for LLM

Delta Compression aims to to reduce the overhead of storing and serving multiple task vectors. Delta compression explores the idea that fine-tuning introduces sparse and structured modifications to a pre-trained model, which can be efficiently compressed. These compressed deltas, when combined with a shared base model, enable storage and inference efficiency by avoiding full model duplication.

- **BitDelta\***. (Liu et al., 2024a) proposes a post-training method that directly quantizes finetuned deltas to 1-bit precision. This strategy reduces both storage and latency while maintaining acceptable fidelity.

- **DeltaCome\***. (Ping et al., 2024) extends delta compression by first applying singular value decomposition (SVD) to each delta and then employing varying bit-widths quantization for different singular vectors based on their singular values.

- **ASVD\***. (Yuan et al., 2023) introduces Activation-aware SVD, a training-free SVD method that improves decomposition accuracy by transforming weight matrices using activation outlier statistics. Although originally proposed for backbone compression, we adapt ASVD to delta compression in our evaluation.

In our experiments, all delta compression baselines are built using the same base model and target finetuned models through SkillWeave, ensuring a controlled comparison.

### D.5 QUANTIZATION FOR LLM

Quantization aims to reduce the bitwidth of model parameters and activations to improve inference speed and reduce memory usage. While weight-only quantization offers moderate savings, full quantization—compressing both weights and activations—is essential to eliminate runtime decompression and unlock true speedups on hardware accelerators. One major challenge in quantization is handling outliers—activation values that are orders of magnitude larger than the rest. These outliers distort the dynamic range and lead to large quantization errors, especially when misaligned with quantization axes. Recent research has focused on outlier-aware quantization to preserve accuracy while enabling aggressive bit reduction.

- **GPTQ** (Frantar et al., 2022) is a post-training quantization method that minimizes quantization error by greedily adjusting non-quantized parameters. It is particularly well-suited for LLMs and supports weight-only quantization.

- **GPTZip** (Isik et al., 2023) extends GPTQ to finetuned deltas, allowing the same quantization process to compress skill-specific updates.

- **LLM.int8** (Dettmers et al., 2022) introduces an 8-bit weight-only quantization framework that use vector-wise quantization to quantize most of the features and isolates the outlier feature dimensions into a 16-bit matrix multiplication for the emergent outliers.

- **AWQ** (Lin et al., 2024) focuses on identifying and protecting salient weight channels using activation outlier statistics. By scaling these channels pre-quantization, AWQ avoids expensive mixed-precision inference and retains performance with pure INT8 computation.

- **SmoothQuant** (Xiao et al., 2023) proposes a full INT8 quantization (W8A8) technique that migrates quantization difficulty from activations to weights via mathematically equivalent transformations.

### D.6 MULTI-TEACHER DISTILLATION

Multi-Teacher Distillation extends classical model distillation by allowing a student model to simultaneously learn from multiple teacher models. Rather than assuming any single teacher is universally superior, this approach aims to distill specialized capabilities from each teacher—leveraging their complementary strengths to form a more holistic student. Recent methods in this line of work typically combine supervised fine-tuning (SFT) with preference-based learning objectives such as Direct Preference Optimization (DPO) or WRPO (Yang et al., 2024b). This hybrid strategy enables the student model to mimic helpful teacher behaviors while suppressing harmful self-generations, promoting both alignment and robustness.

- **FuseLLM\*** (Wan et al., 2024a) is the first to introduce multi-teacher distillation for fusing knowledge from heterogeneous large language models of different scales and structures.

- **FuseChat2.0** (Wan et al., 2024b) refines this idea by a statistics-based token alignment for compatibility. It uses lightweight pairwise fine-tuning into target models of the same size and merges the targets in parameter space.

- **FuseChat3.0\*** (Yang et al., 2025) further introduces implicit model fusion and a DPO-based strategy to enhance alignment and integration performance across heterogeneous LLMs.

We faithfully reproduce these baselines using the official FuseChat-3.0 implementation available at SLIT-AI/FuseChat-3.0. Although our experimental domains differ from those in the original works, we strictly reuse the same teacher models and student architecture to ensure maximum fidelity in reproduction.

# E  EXPERIMENT DETAILS

## E.1  EVALUATION BENCHMARKS

**AlpacaEval-2** (Li et al., 2023) evaluates instruction-following ability using 805 prompts from five datasets, measured by raw win rate (WR) (Dubois et al., 2024) and length-controlled win rate (LC). GPT-4-Preview-1106 serves as both the reference and judge; we report WR in main text and LC in Appendix.

**IFEval** (Zhou et al., 2023) *(Strict, O shot, CoT)* assesses LLMs with automatically verifiable instructions, such as length constraints or required keywords. Evaluation is performed via rule-based parsing, enabling scalable and objective instruction-following assessment.

**GSM8K** (Cobbe et al., 2021)*(O shot, CoT)* is a set of grade-school math word questions that evaluates mathematical reasoning capabilities.

**MATH** (Hendrycks et al., 2021)*(O shot, CoT)* is a dataset of math problems ranging in difficulty from middle school to high school competition level. It tests a wide range of mathematical skills, including algebra, calculus, number theory, and probability.

**HumanEval** (Chen et al., 2021)*(O shot, CoT)* evaluates code generation capabilities by presenting models with function signatures and docstrings and requiring them to implement the function body in Python.

**MBPP** (Austin et al., 2021)*(O shot, CoT)* is a dataset of simple programming problems designed to assess the ability of models to generate short Python code snippets from natural language descriptions.

**BBH** (Suzgun et al., 2023)*(1 shot, CoT)* (Big Bench Hard) targets multi-step logical reasoning and compositional generalization through 23 hand-crafted tasks under few-shot settings.

**ARC-C** (Bhakthavatsalam et al., 2021)*(O shot, CoT)* contains challenging science multiple-choice questions filtered to exclude retrieval or co-occurrence-based solutions, promoting higher-order QA reasoning.

**AgentBench** (Liu et al., 2024b) evaluates agentic reasoning across diverse interactive tasks in executable environments. Each task measures success rate or step success rate under ReAct-style prompting.

## E.2  TRAINING DATASETS

We construct our training set from diverse sources covering a broad spectrum of skills, domains, and instruction styles. The dataset includes both human-written and model-generated examples and overlaps significantly with FUSECHAT-MIXTURE (Wan et al., 2024b).

The full list of training data sources and construction methods is as follows:

- **Math**: OpenMathInstruct-2 [3], MetaMathQA [4], AMC 23 [5]

---

[3] https://huggingface.co/datasets/nvidia/OpenMathInstruct-2
[4] https://huggingface.co/datasets/meta-math/MetaMathQA
[5] https://huggingface.co/datasets/AI-MO/aimo-validation-amc

- **Code**: Self-Oss-Instruct-SC2 [6], OSS-Instruct [7], Evol-Alpaca [8], Python-Code [9].
- **Dialogue**: Magpie-Pro-DPO [10], **Orca-Best**[11], **Capybara**[12], UltraFeedback [13], HelpSteer2 [14], HelpSteer [15], ShareGPT-GPT4 [16].
- **Reasoning**: ARC [17], BBH [18].
- **Agentic**: AgentBench [19], Mind2Web [20], WebShop [21], AgentInstruct (Mitra et al., 2024), AgentBoard [22].

### E.3 HYPERPARAMETER SETTINGS

In DPO experiments, we utilize the TRL library[23] as the training framework for online DPO. We train the LLMs using a batch size of 128 and a maximum length of 4096 on a single node with 8x80GB NVIDIA A800 GPUs. wW use AdamW optimizer with $\beta_1 = 0.9$ and $\beta_2 = 0.999$, weight decay=0.1 with cosine decay and warmup ratio 0.03, BF16 mixed precision, gradient-norm clip 1.0. We truncate sequences to a context length 4096 tokens (prompt+response); padding is left-aligned.

For online sampling (training data generation) we use vLLM with temperature=0.7, top-0.95 p=0.95, top-k=50, n=64 candidates per prompt, max new tokens=2048, and a mild repetition penalty=1.05. At test time, we use greedy decoding (temperature=0)

Because we train separately per task, we select task-specific hyperparameters (*e.g.,* learning rate, epochs, $\beta$. The hyperparameter configurations for different tasks are detailed in Tab. 8. In one word, we adopt a simple yet robust tuning strategy as a general rule:

- For tasks prone to destabilizing training, we reduce the learning rate.
- For inherently harder tasks, we increase the number of online rounds.
- For task where synthetic samples exhibit large distributional variance, we employ a higher $\beta$.

Because all preference pairs are self-generated, the reference term $\log \pi_{ref}(y|x)$ has limited regularization value; we therefore adopt a LN-style (Meng et al., 2024) objective (i.e., DPO with length normalization and without an explicit reference log-ratio)

## F LIMITATION AND FUTURE WORK

While our proposed Skill Weaving framework demonstrates strong performance, parameter efficiency, and generality across diverse domains, we acknowledge several limitations that offer opportunities for further research.

---

[6]https://huggingface.co/datasets/bigcode/self-oss-instruct-sc2-exec-filter-50k
[7]https://huggingface.co/datasets/ise-uiuc/Magicoder-OSS-Instruct-75K
[8]https://huggingface.co/datasets/theblackcat102/evol-codealpaca-v1
[9]https://huggingface.co/datasets/ajibawa-2023/Python-Code-23k-ShareGPT
[10]https://huggingface.co/datasets/Magpie-Align/Magpie-Llama-3.1-Pro-DPO-100K-v0.1
[11]https://huggingface.co/datasets/shahules786/orca-best
[12]https://huggingface.co/datasets/LDJnr/Capybara
[13]https://huggingface.co/datasets/princeton-nlp/llama3-ultrafeedback-armorm
[14]https://huggingface.co/datasets/nvidia/HelpSteer2
[15]https://huggingface.co/datasets/nvidia/HelpSteer
[16]https://huggingface.co/datasets/shibing624/sharegpt_gpt4
[17]https://huggingface.co/datasets/allenai/ai2_arc
[18]https://github.com/suzgunmirac/BIG-Bench-Hard
[19]https://github.com/THUDM/AgentBench
[20]https://github.com/OSU-NLP-Group/Mind2Web
[21]https://github.com/princeton-nlp/webshop
[22]https://github.com/hkust-nlp/AgentBoard
[23]https://github.com/huggingface/trl

Table 8: Hyperparameters for various tasks on Llama-3.1-8B-Instruct model during online DPO stages.

| Target Task | Epochs | DPO Learning Rate | DPO $beta$ | DPO Loss Type |
|---|---|---|---|---|
| Math | 3 | $1 \times 10^{-6}$ | $10 \sim 12$ | $\mathcal{L}_{\text{LN-DPO}}$ |
| Coding | 5 | $5 \times 10^{-7}$ | $10 \sim 12$ | $\mathcal{L}_{\text{LN-DPO}}$ |
| Dialogue | 5 | $1 \times 10^{-6}$ | $5 \sim 8$ | $\mathcal{L}_{\text{LN-DPO}}$ |
| Reasoning | 3 | $9 \times 10^{-7}$ | 8 | $\mathcal{L}_{\text{LN-DPO}}$ |

First, the current construction of SkillPacks relies on human-defined task domains. Although such domain decomposition is intuitive and practical in many cases, it may not fully capture the shared and transferable capabilities across tasks with fuzzy or overlapping boundaries. For example, tasks involving reasoning and dialogue often share latent sub-skills (e.g., commonsense, planning) that cannot be neatly assigned to a single domain. In such cases, fixed domain-level supervision may miss cross-cutting competencies. A promising future direction is to explore automatic skill discovery from large-scale multi-task data, enabling more granular and compositional skill modularization.

Second, our self-improvement pipeline builds upon rule-based verification tailored for each domain, ensuring high-quality automatic feedback. While effective for structured tasks such as code generation or math reasoning, this verification paradigm becomes challenging in open-ended tasks—e.g., creative writing or abstract conversation—where no clear correctness criteria exist. In such scenarios, although other components of our pipeline remain applicable (e.g., skill extraction, skillzip), the lack of explicit verifiability may hinder iterative refinement. Future work may explore hybrid verification strategies that incorporate self-rewarding mechanisms, confidence-based uncertainty estimation, or even reward model scoring, to extend self-improvement to broader task types.

Despite these limitations, our framework already lays a solid foundation for modular, scalable, and self-improving language models. We believe that by addressing the challenges of automatic skill discovery and verification in open-ended settings, Skill Weaving can be further evolved into a general-purpose capability engine adaptable to diverse real-world applications.

