# OpenReview forum: "Skill Weaving: Efficient Self-Improvement of LLMs via Modular Skillpacks"
_ICLR.cc/2026/Conference — ICLR 2026 Conference Withdrawn Submission_

### Official Review · Reviewer_TRFg · 2025-10-30

**Soundness:** 3
**Presentation:** 3
**Contribution:** 2
**Rating:** 4
**Confidence:** 3

**Summary:**

This paper proposes SkillWeave, a modular self-improvement pipeline for LLMs. The idea is to (i) partition a base model’s capabilities into domains, train full-parameter task vectors with online DPO on self-generated data filtered by lightweight rules, then (ii) compress each task vector into an inference-friendly skillpack using SkillZip, a “full quantization” delta-compression scheme that quantizes both low-rank deltas and activations and avoids runtime dequantization. The framework also merges shared knowledge back into a common backbone to reduce interference and memory, and optionally routes tokens to skillpacks at inference (Fig. 1, p. 3). On general-capability benchmarks (math, coding, dialogue, reasoning), the method reports consistent gains over model-merging, LoRA-MoE, and self-improvement baselines (Table 1, p. 7). On agentic tasks (AgentBench), a 7B backbone + ~0.5B skillpacks per domain approaches a 32B monolith while delivering up to ~4× lower latency (Appendix C/Fig. 4).

**Strengths:**

•	Originality / Design coherence. Clear story and easy to understand: extract full-capacity task vectors with preference learning, then compress into efficient skillpacks. The “full-tune-then-zip” position is well-motivated vs. PEFT-only adapters (§3, Fig. 1).
•	Technical quality. SkillZip is thoughtfully engineered: constrained scaling to match tensor-core kernels, channel-wise smoothing (SmoothQuant-like) + rank-wise rotation after SVD, and a concatenation trick for larger GEMMs (§3.2, B.2; Table 2).
•	Empirical scope. Evaluations span four general domains with multiple benchmarks and an agent setting; comparisons include model-merging, routing-MoE (LoRA-MoE/Twin-Merging), self-rewarding, and delta-compression baselines (Table 1; Appendix C).
•	Practical significance. The deployment argument is compelling: shared 7–9B backbone + compact skillpacks, with batch-friendly S-LoRA/VSGGEMM and reported large latency wins (Fig. 3/4 & App.).

**Weaknesses:**

1.	“Self-improvement only” claim vs. data usage. The main text emphasizes improvement “without external labels or stronger teachers,” yet Appendix E.2 lists many human-labeled datasets (e.g., OpenMathInstruct-2, ARC, HumanEval/MBPP sources) used in training/verification, and B.1 uses external reward models for dialogue scoring. Please reconcile what proportion of supervision is truly self-generated vs. derived from labeled corpora or reward models, and how results change if those are removed.
2.	Fairness and budgets. It’s unclear whether baselines received comparable compute and online sampling (e.g., 64 candidates/prompt, long contexts; E.3) and whether they were allowed router/merging hyper-sweeps of similar depth. Report wall-clock, FLOPs, and GPU hours per method/domain.
3.	Statistical reliability. No confidence intervals, seed variability, or bootstrap CIs are reported. Given modest margins on some benchmarks, include variation across ≥3 seeds and, for coding/math, pass@k and solve-rate breakdowns.
4.	Latency reporting inconsistency. The text claims “0.38× speedup over DeltaCome” and “0.04× speedup over S-LoRA” (p. 9). Speedups should be >1 if faster; otherwise report latency ratios (<1). Clarify units (ms/token vs. tokens/s), batch sizes, and kernel vs. end-to-end measurements.
5.	Router and domain inference. Inference-time dispatch is central (Fig. 1 bottom), yet the router (training data, features, losses, OOD behavior) is only briefly mentioned. Provide details and an ablation for mis-routing rates and robustness to mixed-domain prompts.
6.	Interference/forgetting evidence. The paper argues skillpacks “avoid cross-task interference,” but no explicit cross-contamination tests are shown (e.g., train skillpacks sequentially and measure performance decay; probe cross-domain robustness). Add such diagnostics.
7.	Compression fidelity knobs. Truncating INT32 to INT8 between GEMMs (B.2) could be brittle. Include sensitivity to truncation thresholds, rank R, and bit-widths (Xk/Ak/Bk), and compare against weight-only W8A16 + SmoothQuant on the same deltas.  ￼
8.	Writing polish. Several typos/grammar issues and cross-ref errors (e.g., “Tab. 3” in §5.1; inconsistent capitalization of SKillZip). A pass for clarity would help.

**Questions:**

1.	Data provenance: Precisely quantify the fraction of self-generated pairs used in DPO per domain vs. instances using ground-truth labels or external reward models. Can you reproduce Table 1 with only self-generated pairs and rule checks (no reward models, no gold labels)?
2.	Compute and cost: What are the GPU hours for (a) skillpack building and (b) SkillZip per domain? How do these compare to the best-performing baseline families?
3.	Router details: What signals are routed on (prompt tags, token embeddings, a small classifier)? What is the OOD fallback when the input mixes domains? Provide accuracy vs. mis-routing curves.
4.	Interference tests: Can you report sequential skillpack training with periodic re-evaluation on earlier domains to directly evidence reduced forgetting?
5.	Latency metrics: Please reconcile the “× speedup” phrasing and provide a standardized latency table: tokens/s and ms/token at batch sizes {1, 8, 32}, seq-lengths {1k, 4k}, on A100-80G.
6.	Ablations: (i) remove rank-wise rotation; (ii) vary truncation INT32→INT8; (iii) swap DPO for SimPO; (iv) compare against a PEFT-only pipeline with the same SkillZip (i.e., zip LoRA deltas) to isolate the value of full-tune.

---

> ### Author Response · Authors · 2025-11-24
> **Rebuttal - Part I (1/7)**
>
> > ### Question: I think the paper needs to explain more details on how does Skillweave route input token to different skillpacks. This seems to be an important technical part of the proposed method but remains unexplained.
>
> We sincerely **thank the reviewer for this insightful and constructive question.**
> We fully agree that the routing mechanism is an essential technical component of SkillWeave, and we appreciate the reviewer’s careful attention to this aspect.
> We have now **substantially expanded the experiments** of routing and added **a new section “Routing Implementation”** in the Appendix, which provides detailed descriptions, routing model training procedures, quantitative routing accuracy, and inference-time analysis. Below we summarize the key information introduced in the updated manuscript.
>
> ---
>
> ## 1. Clarifying the original routing setup.
>
> **For agent-based tasks**, the task type is inherently known because each tool invocation explicitly specifies its function, and each skillpack corresponds to one such tool-specific or capability-specific domain; therefore **routing is deterministic in real deployments.**
>
> **For general tasks in Table 1**, we now consider two settings:
>  - (1) assumes the task type is already known and the system can activate the correct skillpack according to oracle prompt tags.
> -  (2) **using a lightweight routing model** (finetuned Qwen2.5-0.5B with a linear classification head)  to identify the task domain at inference time.
>
> ---
>
> ## 2. Empirical analysis of the routing mechanism
>
> ## 2.1 Extremely high routing accuracy across domains.
>
> We adopt Qwen2.5-0.5B as a lightweight routing model and attach a linear classification head. The model is fine-tuned for sequence classification using 60,000 labeled prompts across domains. Training is performed for 3 epochs on L40S GPUs, taking approximately one hour.
>
> The routing classifier achieves **>0.999 accuracy/precision/recall** in every domain, with **FPR and FNR <0.002**. These results (Table.3) show that task-type misidentification is exceedingly rare.
>
> Domain | Accuracy |FPR |FNR |Precision |Recall |F1 score  |
> ------ | ------ | ------ | ------ | ------ | ------ | ------   |
> Mathematics |0.9987 |0.0001 |0.0013 |0.9993 |0.9987 |0.9990  |
> Coding      |1.0000 |0.0000 |0.0000 |1.0000 |1.0000 |1.0000 |
> Dialogue    |0.9990 |0.0002 |0.0010 |0.9990 |0.9990 |0.9990  |
> Reasoning   |1.0000 |0.0001 |0.0000 |0.9992 |1.0000 |0.9996  |
>
> ---
>
> ## 2.2 Minimal impact of routing mistakes on downstream performance
>
> We compare oracle routing vs. learned routing (Table.4). The results shows that the performance difference between the two settings is negligible across all benchmarks. This confirms that **routing accuracy is sufficiently high that it does not affect final task performance.**
>
>  |Method | GSM8k | MATH | HumanEval | MBPP | AlpacaEval2 | IFEval | BBH |  ARC-C |
>  |------ | ------ | ------ | ------ | ------ | ------ | ------   |------   |------   |
>  |SKillWeave (**Oracle Routing**) | 91.0 | 62.5 | 75.0 | 77.8 | 52.8 | 79.1 | 76.2 | 88.6 |
>  |SKillWeave (**Learned Routing**) | 91.0 | 62.3 | 75.0 | 77.8 | 52.9 | 78.9 | 76.2 | 88.6 |
>
>
> We manually examined all samples where the router’s prediction differs from the oracle
> domain. Interestingly, **misclassified samples often contain mixed-domain content**, such as a dialogue prompt involving embedded mathematics. We **provide case examples in Appendix B.3**: ``` You extracted 200 milliseconds of sound from a recording with a sampling rate of 48 kHz. How many amplitude measurements do you have now?’’’ ,
> where the “misrouted” “Math” skillpack is actually better suited than the oracle “Dialogue” skillpack.
>
> ---
>
> ## 2.3 Negligible inference-time overhead of routing.
> We further measure the inference-time overhead of routing. As quantified in **Appendix C.3**, routing **increases inference time by <1%** in most settings. The small overhead is **caused by GPU memory contention** rather than routing computation itself, confirming that routing is not a bottleneck.

---

> ### Author Response · Authors · 2025-11-26
> **Rebuttal - Part II (2/7)**
>
> >  ###  Latency reporting inconsistency. Clarify units (ms/token vs. tokens/s), batch sizes, and kernel vs. end-to-end measurements. provide a standardized latency table: tokens/s and ms/token at batch sizes {1, 8, 32}, seq-lengths {1k, 4k}, on A100-80G.
>
>
> We thank the reviewers for the detailed comments regarding the latency evaluation protocol.
> We have significantly expanded the latency section and **added a new Appendix C.3 “Inference Latency Measurement and Results.”** This update substantially clarifies our experimental configurations, provides standardized latency metrics, and presents new measurements across prompt lengths, generation lengths, request rates, and kernel batch sizes.
>
> ---
>
> ## 1. Comprehensive clarification of latency measurement settings
>
> Appendix C.3 now provides a complete description of our end-to-end latency setup, including:
>  - **Hardware**: all experiments on a single NVIDIA A100-80G
> - **Inference engine**: S-LoRA as the primary backend (chosen because it enables direct integration of our optimized low-rank kernels; our method remains compatible with vLLM/sGLang)
> - **Model configuration**: Llama3.1-8B-Instruct as the base model; skillpack matrices with effective rank ≈ 400 (ranging from 300 to 600 depending on the module).
> - **Request arrival process**: The requests for each domain are modeled using a Gamma arrival process with a coefficient of variance of 1Gamma arrival process (CV = 1), which produces realistic bursty workloads and is used consistently across Figures 3 and 5
> - **Reported statistics**: *request throughput (req/s), token throughput (tokens/s), ms/token, and full distribution metrics (mean, median, P95, P99, min, max)*
>
> These details ensure that all latency results are now fully reproducible.
>
> ---
>
> ## 2. Clarifying multi-model baseline deployment
>
> To clarify the 5×7B multi-model baseline  used in Figures 3 and 5, we have added a detailed description in **Appendix C.3** to ensure full transparency. Briefly, each finetuned 7B model is served as an independent **vLLM worker under Nvidia MPS**, with GPU memory proportionally allocated to its expected request rate. When a domain becomes inactive, the corresponding model is evicted and replaced by the next most frequent one, and all model-loading overhead is fully accounted for in the latency numbers.
>
> This deployment strategy is the **most favorable and realistic setting** for multi-model systems on a single A100-80G: the GPU can **host at most 4–5 models simultaneously** before running out of memory, and using fewer models degenerates into sequential reloading with prohibitively high latency. We therefore select N = 4 as the optimal baseline configuration.
>
> We highlight that, even under this best-case deployment for multi-model baselines, SkillWeave consistently achieves substantially lower end-to-end latency because it avoids model swaps entirely and keeps all skillpacks active with lightweight low-rank kernels. We encourage reviewers to compare the full results in Appendix C.3.
>
> ---
>
> ## 3. New experiments across diverse runtime conditions (Table 6)
>
> To make latency behavior more transparent, we added **three new groups of experiments**, covering:
> 1. **Different prompt lengths** (50 / 200 / 500)
> 2. **Different generation lengths** (20 / 100 / 200 /1000)
> 3. **Different request arrival rates** (0.5 → 7 req/s)
> 4. including **long generation cases** (4000 tokens)
>
> These settings produce a broad and realistic latency landscape.
> Across all metrics, SkillWeave consistently achieves lower latency and higher throughput compared with multi-model baselines.
> We encourage reviewers to inspect Table 6 for detailed quantitative results.
>
> ---
>
> ## 4. New kernel-level latency results for different batch sizes (Table 7)
>
> Given that modern inference engines (vLLM, sGLang) use continuous dynamic batching, the actual batch size depends on concurrent traffic rather than user-specified settings. Although reviewers asked for latency under batch sizes {1, 8, 32}, end-to-end batch-size-controlled measurements are therefore not meaningful.
>
> To address this request in a principled way, we:
> - approximated effective batch size via **controlled request concurrency** (following vLLM evaluation protocols) in Table 6,
> - and added a new **kernel-level batch-sweep experiment** (Table 7) using
>     - Llama3.1-8B-Instruct
>     - down_proj matrix (4096×14336), rank=600
>     - prefill seq_len=1000, decode seq_len=1
>     - batch sizes {1, 4, 10, 32}
>
> These results clearly show that our **X8A8A8 low-rank kernel is substantially faster** than the backbone’s FP16 (X16W16) kernel across all batch settings, and the latency overhead introduced by activating a skillpack is minimal.
>
> ---
>
> > ### Please reconcile the “× speedup” phrasing. Speedups should be >1 if faster; otherwise report latency ratios
>
> We sincerely thank the reviewer for this careful and attentive suggestions. We have already addressed this issue in the manuscript

---

> ### Author Response · Authors · 2025-11-26
> **Rebuttal - Part III  (3/7)**
>
> > ### “Self-improvement only” claim vs. data usage. Please reconcile what proportion of supervision is truly self-generated vs. derived from labeled corpora or reward models, and how results change if those are removed.
>
> We sincerely thank the reviewer for raising this important conceptual question. We fully acknowledge that “self-improvement” is a nuanced term in LLM training, especially when verification or filtering mechanisms are involved. Below we clarify our claims, address the reviewer’s concerns point-by-point, and provide **two new controlled experiments** that directly demonstrate SkillWeave’s effect **independent of data advantages.**
>
> ---
>
> ## 1. Acknowledging the limitation (already stated explicitly in the main paper)
> In our Limitations section, we already state that our verification-based self-improvement maybe not applicable to absolute unsupervised self-evolve. We agree this is a real limitation and do not claim universal self-improvement across all task types.
>
> We appreciate the reviewer highlighting this point.  We respectfully request reviewers not to harshly criticize this limitation which is important but not disqualifying — as every work has its own limitations.
>
> ---
>
> ## 2. Clarifying what “self-improvement” means in SkillWeave
> ## 2.1 From the perspective of data usage
> **All training data used for SkillWeave’s refinement process are self-generated** —
> we do **not** use human-written solutions or stronger-teacher outputs to supervise training.
> Thus, the source of preformance gain is ultimately the model’s own capabilities.
> Even if verification is not “fully unsupervised,” the improvement signal comes from the model’s own outputs—hence our use of “self-improvement.”
>
> ## 2.2 From the perspective of method design
>
> SkillWeave’s self-improvement does not rely solely on self-generated data.
> Its key innovation is an architectural and training paradigm that: **decompose the LLM’s overall capacity into skillpacks and isolates training by domain, followed by compact compression.**
>
> Even if all methods were trained on the exact same dataset, SkillWeave still improves over the base model—because **the improvement arises from reorganizing the model’s internal structure, not only from privileged data.**
>
> To illustrate this point explicitly, we conducted two new sets of controlled experiments.
>
> ---
>
> ## 3. New controlled experiments (strictly equal data and model size): SkillWeave remains the best
>
> To directly address the reviewer's concern, we conducted **new experiments** where:
>  -  **All methods are trained on identical self-generated DPO data**
>  -  **All methods use the same model size: 10B**
>
>
> We compare five strong baselines:
> 1. **PEFT**: per-domain LoRA finetuning
> 2. **Routed-LoRA**
> 3. **ASVD / DeltaCome**: per-domain full finetuning + per-domain delta compression
> 4. **LoRA-MoE**: mixture-of-LoRA training on multi-domain data
> 5. **Twin-Merging**: per-domain full finetuning + low-rank twin merging
>
> Method | GSM8K	| Math	| HumanEval	| MBPP	| AlpacaEval	| IFEval	| BBH	| ARC
> ----|---- |---- |---- |---- |---- | ----|---- | ----
> base		| 84.5	| 51.9	| 69.5	| 75.4	| 28.3	| 75.9	| 65.8	| 82.4
> SKillWeave	| **91.0**	| **62.5**	| **75.0**	| *77.8*	| **52.8**	| **79.1**	| *76.2*	| **88.6**
> →PEFT		| 86.8	| 49.3	| 73.5	| 74.2	| 47.9	| 78.1	| 68.5	| 86.9
> →ASVD		| 89.7	| 60.5	| 73.7	| 76.9	| 47.0	| 78.3	| 74.3	| 87.2
> →DeltaCome	| *90.7*	| *62.4*	| *74.9*	| **78.0**	| *52.7*	| *79.0*	| **76.3**	| *88.4*
> Twin-Merging 	| 87.2	| 58.8	| 72.8	| 75.3	| 46.3	| 77.5	| 71.4	| 86.1
> RoutedLoRA	| 87.4	| 52.0	| 73.7	| 74.3	| 50.4	| 78.7	| 69.2	| 86.7
>
> ### Results:
> Across all domains and metrics, **SkillWeave remains the top-performing method** even under perfectly controlled data conditions. This strongly supports that the gains originate from our architectural design—**the full-tuning–then-zip paradigm that reorganizes and refines internal capabilities**, not from additional labeled data only.

---

> ### Author Response · Authors · 2025-12-02
> **Rebuttal - Part VI  (4/7)**
>
> > ### Interference/forgetting evidence. The paper argues skillpacks “avoid cross-task interference,” but no explicit cross-contamination tests are shown (e.g., train skillpacks sequentially and measure performance decay; probe cross-domain robustness). Add such diagnostics.
>
> We added **new controlled experiment** to further examine whether SkillWeave alleviates interference and forgetting even when all models see the same data. Using the same training set, we compare:
> 1. **Mixed-domain MTL**: all domains merged for full finetuning
> 2. **Continual-learning MTL**: sequential multi-domain full finetuning
> 3. **Model Merging**: per-domain full finetuning + merging
>
> Method | GSM8K	| Math	| HumanEval	| MBPP	| AlpacaEval	| IFEval	| BBH	| ARC
> ----|---- |---- |---- |---- |---- | ----|---- | ----
> SKillWeave	| **91.0**	| **62.5**	| **75.0**	| **77.8**	| **52.8**	| **79.1**	| 76.2	| **88.6**
> Mixed-domain MTL |87.5		|53.7	|70.7	| 75.8	| 37.9	| 77.8	| 67.3	| 83.2
> Continual-learning |86.7 	|52.1	|69.4	| 75.2	| 39.2	| 78.6	| **76.3**	| 88.2
> Model Merging |88.9	| 57		| 71.5		| 76.2		| 35		| 77.2		| 71.2		| 84.6
>
> Our analysis shows that:
>  - Conflict analysis: Skillpacks for different domains exhibit negative sign consistency (−0.15) ,  low cosine similarity (0.29).
> → Multi-domain full finetuning inevitably suffers from task interference and catastrophic forgetting.
> - Shared-knowledge extraction:
> The merged backbone shows high similarity with all task vectors (cosine = 0.153, sign consistency=0.64),
> → showing that model merging identifies generalizable capabilities.
> - Compression becomes easier after merging:
> Mean Absolute Value decreases by ~33%, improving SVD-based compression accuracy.
> → SkillWeave separates general knowledge and conflicting knowledge, enabling stable refinement.
>
> These results show that SkillWeave can alleviate interference and forgetting.

---

### Official Review · Reviewer_2YaN · 2025-11-01

**Soundness:** 3
**Presentation:** 4
**Contribution:** 3
**Rating:** 6
**Confidence:** 2

**Summary:**

The paper proposes SkillWeave, a modular self-improvement framework for LLMs that turns one general model into a set of domain-specialized skillpacks, each trained with the model’s own synthetic data and filtered by lightweight rule-based checks, then optimized by DPO to prefer helpful over harmful generations. Additionally, the paper introduces SkillZip, an inference-oriented delta-compression scheme that first merges shared knowledge into the backbone and then fully quantizes both deltas and activations, enabling direct INT8/INT4 computation without runtime dequantization. The proposed method demonstrates good performance and efficiency on several agentic benchmarks.

**Strengths:**

-  The paper is clearly written and organized.
- The experiments are very comprehensive with comparisons over many related works.
- The experiments demonstrate superior performance showing that a small SkillWeave model trained only on self-generated data can match or exceed the performance of teacher-assisted systems while being faster at inference.

**Weaknesses:**

- Some typos in Section C of the Appendix, where the figure and table indexes are in mismatch.
- The rule-based self verification seems to be unapplicable to more generic skills/tasks.

**Questions:**

- I think the paper needs to explain more details on how does Skillweave route input token to different skillpacks. This seems to be an important technical part of the proposed method but remains unexplained.

---

> ### Author Response · Authors · 2025-11-24
> **Rebuttal - Part I  (1/2)**
>
> > ### Question: I think the paper needs to explain more details on how does Skillweave route input token to different skillpacks. This seems to be an important technical part of the proposed method but remains unexplained.
>
> We sincerely **thank the reviewer for this insightful and constructive question.**
> We fully agree that the routing mechanism is an essential technical component of SkillWeave, and we appreciate the reviewer’s careful attention to this aspect.
> We have now **substantially expanded the experiments** of routing and added **a new section “Routing Implementation”** in the Appendix, which provides detailed descriptions, routing model training procedures, quantitative routing accuracy, and inference-time analysis. Below we summarize the key information introduced in the updated manuscript.
>
> ---
>
> ## 1. Clarifying the original routing setup.
>
> **For agent-based tasks**, the task type is inherently known because each tool invocation explicitly specifies its function, and each skillpack corresponds to one such tool-specific or capability-specific domain; therefore **routing is deterministic in real deployments.**
>
> **For general tasks in Table 1**, we now consider two settings:
>  - (1) assumes the task type is already known and the system can activate the correct skillpack accordingly.
> -  (2) **using a lightweight routing model** (finetuned Qwen2.5-0.5B with a linear classification head)  to identify the task domain at inference time.
>
> ---
>
> ## 2. Empirical analysis of the routing mechanism
>
> ## 2.1 Extremely high routing accuracy across domains.
>
> We adopt Qwen2.5-0.5B as a lightweight routing model and attach a linear classification head. The model is fine-tuned for sequence classification using 60,000 labeled prompts across domains. Training is performed for 3 epochs on L40S GPUs, taking approximately one hour.
>
> The routing classifier achieves **>0.999 accuracy/precision/recall** in every domain, with **FPR and FNR <0.002**. These results (Table.3) show that task-type misidentification is exceedingly rare.
>
> Domain | Accuracy |FPR |FNR |Precision |Recall |F1 score  |
> ------ | ------ | ------ | ------ | ------ | ------ | ------   |
> Mathematics |0.9987 |0.0001 |0.0013 |0.9993 |0.9987 |0.9990  |
> Coding      |1.0000 |0.0000 |0.0000 |1.0000 |1.0000 |1.0000 |
> Dialogue    |0.9990 |0.0002 |0.0010 |0.9990 |0.9990 |0.9990  |
> Reasoning   |1.0000 |0.0001 |0.0000 |0.9992 |1.0000 |0.9996  |
>
> ---
>
> ## 2.2 Minimal impact of routing mistakes on downstream performance
>
> We compare oracle routing vs. learned routing (Table.4). The results shows that the performance difference between the two settings is negligible across all benchmarks. This confirms that **routing accuracy is sufficiently high that it does not affect final task performance.**
>
>  |Method | GSM8k | MATH | HumanEval | MBPP | AlpacaEval2 | IFEval | BBH |  ARC-C |
>  |------ | ------ | ------ | ------ | ------ | ------ | ------   |------   |------   |
>  |SKillWeave (**Oracle Routing**) | 91.0 | 62.5 | 75.0 | 77.8 | 52.8 | 79.1 | 76.2 | 88.6 |
>  |SKillWeave (**Learned Routing**) | 91.0 | 62.3 | 75.0 | 77.8 | 52.9 | 78.9 | 76.2 | 88.6 |
>
>
> We manually examined all samples where the router’s prediction differs from the oracle
> domain. Interestingly, **misclassified samples often contain mixed-domain content**, such as a dialogue prompt involving embedded mathematics. We **provide case examples in Appendix B.3**: ``` You extracted 200 milliseconds of sound from a recording with a sampling rate of 48 kHz. How many amplitude measurements do you have now?’’’ ,
> where the “misrouted” “Math” skillpack is actually better suited than the oracle “Dialogue” skillpack.
>
> ---
>
> ## 2.3 Negligible inference-time overhead of routing.
> We further measure the inference-time overhead of routing. As quantified in **Appendix C.3**, routing **increases inference time by <1%** in most settings. The small overhead is **caused by GPU memory contention** rather than routing computation itself, confirming that routing is not a bottleneck.
>
> ---
>
> ## 3. Domain-level routing design choices
>
> We also clarify that SkillWeave does not support token-level MoE-style expert routing, due to (i) large hidden-state divergence across independently trained skillpacks and (ii) The KV cache between multiple Skillpacks is internally continuous and externally isolated in virtual memory，and the inference engine of MoE is completely different. We acknowledge this as a current limitation and plan to explore improved designs in future work.

---

> ### Author Response · Authors · 2025-11-26
> **Rebuttal - Part II   (2/2)**
>
> > ### The rule-based self verification seems to be unapplicable to more generic skills/tasks.
>
> We sincerely thank the reviewer for this thoughtful observation. We completely agree that rule-based or programmatic self-verification is naturally more applicable to structured domains such as code or math, and may not directly generalize to open-ended tasks. Below we provide **two new controlled experiments** emphasizing that **the performance gains of SkillWeave arise primarily from its architectural design rather than from verification alone.**
>
> ---
>
> ## 1. Acknowledging the limitation (already stated explicitly in the main paper)
> We appreciate the reviewer’s accurate reading of the paper.
> In our Limitations section, we explicitly note that rule-based verification is most effective for tasks with clear correctness criteria, and its applicability becomes limited for creative generation, dialogue, or abstract reasoning. We fully agree with the reviewer that this is an important boundary, and we do not claim universal applicability of verification-based self-refinement.
>
> ---
>
> ## 2. SkillWeave’s self-improvement is not primarily driven by verification signals
> While rule-based self-verification helps filter model-generated data in some domains, the core contribution of SkillWeave lies in its architectural and training paradigm: *decompose the LLM’s overall capacity into skillpacks and isolates training by domain, followed by compact compression.*
> Verification is a helpful component, but not the primary driver of SkillWeave’s improvements.
> To directly support this clarification, we conducted two new controlled experiments.
>
>
> ---
>
> ## 3. New controlled experiments (strictly equal data and model size): SkillWeave remains the best
> To isolate SkillWeave’s design contribution from its verification component, we ran **new experiments** where:
>  -  **All methods are trained on identical self-generated DPO data**
>  -  **All methods use the same model size: 10B**
>
>
> We compare five strong baselines:
> 1. **PEFT**: per-domain LoRA finetuning
> 2. **Routed-LoRA**
> 3. **ASVD / DeltaCome**: per-domain full finetuning + per-domain delta compression
> 4. **LoRA-MoE**: mixture-of-LoRA training on multi-domain data
> 5. **Twin-Merging**: per-domain full finetuning + low-rank twin merging
>
> Method | GSM8K	| Math	| HumanEval	| MBPP	| AlpacaEval	| IFEval	| BBH	| ARC
> ----|---- |---- |---- |---- |---- | ----|---- | ----
> base		| 84.5	| 51.9	| 69.5	| 75.4	| 28.3	| 75.9	| 65.8	| 82.4
> SKillWeave	| **91.0**	| **62.5**	| **75.0**	| *77.8*	| **52.8**	| **79.1**	| *76.2*	| **88.6**
> →PEFT		| 86.8	| 49.3	| 73.5	| 74.2	| 47.9	| 78.1	| 68.5	| 86.9
> →ASVD		| 89.7	| 60.5	| 73.7	| 76.9	| 47.0	| 78.3	| 74.3	| 87.2
> →DeltaCome	| *90.7*	| *62.4*	| *74.9*	| **78.0**	| *52.7*	| *79.0*	| **76.3**	| *88.4*
> Twin-Merging 	| 87.2	| 58.8	| 72.8	| 75.3	| 46.3	| 77.5	| 71.4	| 86.1
> RoutedLoRA	| 87.4	| 52.0	| 73.7	| 74.3	| 50.4	| 78.7	| 69.2	| 86.7
>
> ### Results:
> Across all domains —including those without programmatic verification—SkillWeave consistently outperforms all baselines. The improvements remain even when the data advantage of verification is removed.
>
> ---
>
> ## 4. New controlled experiment: SkillWeave alleviates interference & forgetting (higher data efficiency)
> We further examine why SkillWeave improves even when all models see the same data. Using the same training set, we compare:
> 1. **Mixed-domain MTL**: all domains merged for full finetuning
> 2. **Continual-learning MTL**: sequential multi-domain full finetuning
> 3. **Model Merging**: per-domain full finetuning + merging
>
> Method | GSM8K	| Math	| HumanEval	| MBPP	| AlpacaEval	| IFEval	| BBH	| ARC
> ----|---- |---- |---- |---- |---- | ----|---- | ----
> SKillWeave	| **91.0**	| **62.5**	| **75.0**	| **77.8**	| **52.8**	| **79.1**	| 76.2	| **88.6**
> Mixed-domain MTL |87.5		|53.7	|70.7	| 75.8	| 37.9	| 77.8	| 67.3	| 83.2
> Continual-learning |86.7 	|52.1	|69.4	| 75.2	| 39.2	| 78.6	| **76.3**	| 88.2
> Model Merging |88.9	| 57		| 71.5		| 76.2		| 35		| 77.2		| 71.2		| 84.6
>
> Our analysis shows that:
>  - Conflict analysis: Skillpacks for different domains exhibit negative sign consistency (−0.15) ,  low cosine similarity (0.29).
> → Multi-domain full finetuning inevitably suffers from task interference and catastrophic forgetting.
> - Shared-knowledge extraction:
> The merged backbone shows high similarity with all task vectors (cosine = 0.153, sign consistency=0.64),
> → showing that model merging identifies generalizable capabilities.
> - Compression becomes easier after merging:
> Mean Absolute Value decreases by ~33%, improving SVD-based compression accuracy.
> → SkillWeave separates general knowledge and conflicting knowledge, enabling stable refinement.
>
> These results show that SkillWeave’s gains arise from more effective use of the same data, rather than solely depending on verification-specific signals.
>
> ---
>
> ### We hope these clarifications are helpful, and we really appreciate the reviewer’s careful reading and constructive feedback.

---

### Official Review · Reviewer_3hJJ · 2025-11-01

**Soundness:** 3
**Presentation:** 2
**Contribution:** 3
**Rating:** 4
**Confidence:** 3

**Summary:**

SkillWeave is a modular self-improvement framework for large language models that splits a model’s general abilities into smaller, domain-specific “skillpacks.” The framework introduces SkillZip, a fully quantized compression method that reduces both weights and activations via a smooth-quant variant to achieve faster inference. Together, SkillWeave and SkillZip enable efficient, self-supervised specialization without extra human labels. Experiments show a the collection of the 5x7B SkillWeave model outperforms 32B monolithic LLMs while running up to four times faster.

**Strengths:**

1. Modularity: SkillWeave effectively decomposes large language models into domain-specific skillpacks that can be trained, updated, and deployed independently. This modular separation helps reduce cross-task interference and prevents catastrophic forgetting, enabling more stable multi-domain specialization.

2. Hardware-aware compression: The integrated SkillZip module performs full quantization of both weights and activations, allowing computation directly in low-bit formats. This design achieves significant improvements in inference speed and memory efficiency without requiring runtime decompression.

3. Performance-efficiency balance: The benefits of SkillWeave are clearly shown in comparisons between monolithic 32B models and distributed 5×7B model setups, where SkillWeave delivers similar or superior accuracy while achieving faster inference.

**Weaknesses:**

1. Model size comparison: Several baselines in Table 1 use smaller models (e.g., 8B) compared to the 10B SkillWeave configuration. This makes it difficult to clearly separate improvements due to the proposed method from those arising from the increased model capacity. A fairer comparison with models of equal or similar size would strengthen the empirical claims.

2. Overloaded figure presentation: Figure 1 contains many overlapping components, which makes the core pipeline hard to follow. The authors should consider restructuring the figure and introducing each stage -- skillpack construction, compression, and integration -- sequentially in the text to improve clarity and reader comprehension.

3. Need for clearer narrative focus: The writing often shifts between unrelated technical details -- such as quantization smoothing, preference optimization, and inference serving optimizations -- without emphasizing the central contribution. Streamlining the exposition to first establish the main SkillWeave pipeline and deferring secondary engineering details (e.g., quantization specifics, GEMM parallelization) to the appendix would make the paper more coherent and easier to follow. If the authors could provide a more central pipeline of their method, this would aid in distinguishing between engineering/implementation features and central methodology.

**Questions:**

Refer to the requested changes in the Weaknesses Section.

---

> ### Author Response · Authors · 2025-11-24
> **Rebuttal - Part I   (1/3)**
>
> > ### Model size comparison: Several baselines in Table 1 use smaller models (e.g., 8B) compared to the 10B SkillWeave configuration. This makes it difficult to clearly separate improvements due to the proposed method from those arising from the increased model capacity. A fairer comparison with models of equal or similar size would strengthen the empirical claims.
>
> We sincerely thank the reviewer for raising this important concern regarding the model size fairness in Table 1. Below, (1) Clarifying the comparisons in Table 1 (the original comparison is fair)  (2) provide substantial additional experiment that **strictly controlled 10B-vs-10B comparisons under identical data and training settings**
>
> ---
>
> ## 1. Clarifying the comparisons in Table 1 (the original comparison is fair)
>
> **Most baselines in Table 1 actually use models ≥ 10B.**  Across the six categories of baselines:
>
>
> - (1) *Open-source LLMs, Routing-based LLMs, and Our Approach + Optional Replacement*.
> these three categories include **10 baseline systems, all using ≥10B parameters**, with several ranging **from 16.7B to 21B**—substantially larger than the 10B SkillWeave model.
> → Thus, for the majority of baselines, SkillWeave is compared against equal or larger models.
>
> - (2) *Self-Rewarding*.
> We evaluate both 8B and 10B versions of this method. SkillWeave **outperforms both 8B and 10B**:
>     - This shows our improvement cannot be attributed to model scale.
>     - More importantly, it highlights that SkillWeave extracts much more utility from self-generated data, which is the core difficulty of self-improvement settings.
>
> - (3) *Multi-Teacher Distillation*.
> Teacher Distillation uses external data from **stronger teacher models.**
> → This baseline is inherently unfair in favor of the baseline.
> We include it only as **a performance upper bound.**
>
> - (4) *Model Merging & Model Grafting*.
> We agree that these fusion-type baselines *do not scale model capacity*. However, their performance is structurally capped. SkillWeave **raises the performance ceiling for fusion-style methods**, and the moderate parameter increase is a necessary tradeoff for resolving domain conflicts.
>
> - (5) **SkillWeave’s main comparison target is routing-based modular LLMs**.
> SkillWeave is an efficient modular framework. Against the most relevant baselines—*existing routing-based methods*—SkillWeave achieves **Smaller or comparable parameter count  and Higher performance**
>
> ---
>
> ## 2. New controlled experiments (strictly equal model size and data): SkillWeave remains the best
>
> To directly address the reviewer's concern, we conducted **new experiments** where:
>  -  **All methods use the same model size: 10B**
>  -  **All methods are trained on identical self-generated DPO data**
>
> We compare five strong baselines:
> 1. **PEFT**: per-domain LoRA finetuning
> 2. **Routed-LoRA**
> 3. **ASVD / DeltaCome**: per-domain full finetuning + per-domain delta compression
> 4. **LoRA-MoE**: mixture-of-LoRA training on multi-domain data
> 5. **Twin-Merging**: per-domain full finetuning + low-rank twin merging
>
> Method | GSM8K	| Math	| HumanEval	| MBPP	| AlpacaEval	| IFEval	| BBH	| ARC
> ----|---- |---- |---- |---- |---- | ----|---- | ----
> base		| 84.5	| 51.9	| 69.5	| 75.4	| 28.3	| 75.9	| 65.8	| 82.4
> SKillWeave	| **91.0**	| **62.5**	| **75.0**	| *77.8*	| **52.8**	| **79.1**	| *76.2*	| **88.6**
> →PEFT		| 86.8	| 49.3	| 73.5	| 74.2	| 47.9	| 78.1	| 68.5	| 86.9
> →ASVD		| 89.7	| 60.5	| 73.7	| 76.9	| 47.0	| 78.3	| 74.3	| 87.2
> →DeltaCome	| *90.7*	| *62.4*	| *74.9*	| **78.0**	| *52.7*	| *79.0*	| **76.3**	| *88.4*
> Twin-Merging 	| 87.2	| 58.8	| 72.8	| 75.3	| 46.3	| 77.5	| 71.4	| 86.1
> RoutedLoRA	| 87.4	| 52.0	| 73.7	| 74.3	| 50.4	| 78.7	| 69.2	| 86.7
>
>
>
> ### Results:
> Across all domains and metrics, **SkillWeave remains the top-performing method** even under perfectly controlled model size conditions. This strongly supports that the gains originate from our architectural design—**the full-tuning–then-zip paradigm that reorganizes and refines internal capabilities**, rather than increased parameter count.

---

> ### Author Response · Authors · 2025-11-27
> **Rebuttal - Part II  (2/3)**
>
> ---
>
> ## 3.Additional evidence: SkillWeave achieves much higher data efficiency and lifts the performance upper bound  (than parameter-efficient methods)
>
>
> We acknowledge that SkillWeave increases model parameters, but in self-improvement settings, **data efficiency is the true bottleneck.**
> We therefore run additional experiments using the **same self-generated training data** under three settings:
>
> 1. **Mixed-domain MTL**: all domains merged for full finetuning
> 2. **Continual-learning MTL**: sequential multi-domain full finetuning
> 3. **Model Merging**: per-domain full finetuning + merging
>
> Method | GSM8K	| Math	| HumanEval	| MBPP	| AlpacaEval	| IFEval	| BBH	| ARC
> ----|---- |---- |---- |---- |---- | ----|---- | ----
> SKillWeave	| **91.0**	| **62.5**	| **75.0**	| **77.8**	| **52.8**	| **79.1**	| 76.2	| **88.6**
> Mixed-domain MTL |87.5		|53.7	|70.7	| 75.8	| 37.9	| 77.8	| 67.3	| 83.2
> Continual-learning |86.7 	|52.1	|69.4	| 75.2	| 39.2	| 78.6	| **76.3**	| 88.2
> Model Merging |88.9	| 57		| 71.5		| 76.2		| 35		| 77.2		| 71.2		| 84.6
>
> These parameter-efficient baselines **do not improve after a certain point.**
> Their capacity to absorb conflicting cross-domain signals is inherently limited.
>
> Our analysis shows that:
>  - Conflict analysis: Skillpacks for different domains exhibit negative sign consistency (−0.15) ,  low cosine similarity (0.29).
> → Multi-domain full finetuning inevitably suffers from task interference and catastrophic forgetting.
> - Shared-knowledge extraction:
> The merged backbone shows high similarity with all task vectors (cosine = 0.153, sign consistency=0.64),
> → showing that model merging identifies generalizable capabilities.
> - Compression becomes easier after merging:
> Mean Absolute Value decreases by ~33%, improving SVD-based compression accuracy.
> → SkillWeave separates general knowledge and conflicting knowledge, enabling stable refinement.
>
>
> **In short:**
> SkillWeave isolates domain-conflicting parameters into skillpacks and retains domain-general parameters in the shared backbone.
> **This eliminates forgetting and raises the performance ceiling under limited data** than these 8B parameter-efficient baselines
>
> ---
> ---
> ---
>
>
> > ### *Weakness 3*: Need for clearer narrative focus: The writing often shifts between unrelated technical details -- such as quantization smoothing, preference optimization, and inference serving optimizations -- without emphasizing the central contribution. ........
>
>
> We sincerely thank the reviewer for this thoughtful and constructive suggestion. We fully agree that stronger narrative focus and a clearer separation between core methodology and engineering details would significantly improve the readability of the paper. In response, we have **made substantial revisions to the main text.**
>
> **1. We restructured the beginning of the Method section to provide a clearer, more focused pipeline overview.**
> We added a dedicated introductory subsection that now:
>  - clearly defines the problem setting and objectives of SkillWeave,
>  - outlines the full three-stage pipeline (Skillpack Building → Skillpack Compression → Skillpack Integration), and
>  - explains how these stages constitute the central methodological contribution.
>
> This new subsection provides the high-level narrative flow you suggested and ensures that readers grasp the conceptual framework before encountering any technical detail.
>
> **2. We streamlined the main text by removing secondary implementation details.**
>
> Following your recommendation, we moved several engineering-oriented components out of the core exposition and into the Appendix, including: parameter concatenation, inference-engine implementation notes and kernel-level optimizations.
> This significantly declutters the main narrative and keeps the method section centered on the algorithmic pipeline.
>
>
> We truly appreciate the reviewer’s insight; your feedback has led to a substantially improved paper. We will continue refining the presentation to keep the core contributions as clear and accessible as possible.

---

### Official Review · Reviewer_osN7 · 2025-11-03

**Soundness:** 3
**Presentation:** 3
**Contribution:** 2
**Rating:** 6
**Confidence:** 4

**Summary:**

This work introduces lightweight, domain-specific skillpacks that boost domain performance for LLMs. It beats other baselines on most domain related benchmarks while using self-generated data without relying on costly human labels or stronger teachers. For deployment, the paper introduces SkillZip, a quantization strategy that  packages domain skillpacks in a fully quantized form. It jointly quantizes weights and activations to avoid runtime decompression/dequantization and reduce latency. SkillWeave  and SkillZip as a unified pipeline enables self-improvement and efficient deployment for LLMs.

**Strengths:**

1. Clear accuracy gains with strong baselines. It consistently matches or exceeds competitive merging, routing, and self-improvement baselines of comparable or larger size.

2. Hardware-aware design with strong throughput. The agentic evaluation uses a single 7B backbone plus five 0.5B skillpacks and shows 4.2× faster inference than a 32B monolithic model and 5.5× faster than deploying five separate 7B models , while staying within 3–5% of their accuracy. The SkillZip study further indicates that fully quantized deltas (e.g., X8A8B8, X8A4B4) maintain or slightly exceed DeltaCome’s accuracy at lower latency.

**Weaknesses:**

1. In Table 1, the reported accuracy assumes the task type is already known and the system can activate the correct skillpack accordingly. However, how to reliably infer the task type at inference time remains unspecified. Without quantified routing accuracy (misclassification rate,  fallback), it is unclear how much task-type misidentification would hurt downstream accuracy.

2. Critical E2E inference settings (prefill & decode length, batch size) are missing for Figure 3.

**Questions:**

1. Can you clarify on how to use the 5x7B model for domain specific tasks? Is model loading time also included in the inference latency?

2. In section 5.1, you metioned a 7B backbone but Table 1 uses 8B backbone.

---

> ### Author Response · Authors · 2025-11-24
> **Rebuttal - Part I   (1/3)**
>
> > ### In Table 1, the reported accuracy assumes the task type is already known and the system can activate the correct skillpack accordingly. However, how to reliably infer the task type at inference time remains unspecified.  it is unclear how much task-type misidentification would hurt downstream accuracy.
>
> We sincerely thank the reviewers for raising this important concern regarding SkillWeave’s routing mechanism. We agree that assuming known task types is overly idealized, and in response, we have conducted **substantial additional experiments** and added **a new section “Routing Implementation”** in the Appendix to fully clarify this component. Below we summarize how these new results directly address the reviewers’ questions.
>
> ---
>
> ## 1. Clarifying the original routing setup.
>
> **For agent-based tasks**, the task type is inherently known because each tool invocation explicitly specifies its function, and each skillpack corresponds to one such tool-specific or capability-specific domain; therefore **routing is deterministic in real deployments.**
>
> **For general tasks in Table 1**, the previous assumption of known domains is indeed a limitation. To address this, **we now train a dedicated routing model** (Qwen2.5-0.5B with a linear head) that predicts domain labels at inference time.
>
> ---
>
> ## 2. Strong new evidence that routing does not affect SkillWeave’s performance.
>
> ## 2.1 Routing accuracy is extremely high.
>
> We adopt Qwen2.5-0.5B as a lightweight routing model and attach a linear classification head. The model is fine-tuned for sequence classification using 60,000 labeled prompts across domains. Training is performed for 3 epochs on L40S GPUs, taking approximately one hour.
>
> The routing classifier achieves **>0.999 accuracy/precision/recall** in every domain, with **FPR and FNR <0.002**. These results (Table.3) show that task-type misidentification is exceedingly rare.
>
> Domain | Accuracy |FPR |FNR |Precision |Recall |F1 score  |
> ------ | ------ | ------ | ------ | ------ | ------ | ------   |
> Mathematics |0.9987 |0.0001 |0.0013 |0.9993 |0.9987 |0.9990  |
> Coding      |1.0000 |0.0000 |0.0000 |1.0000 |1.0000 |1.0000 |
> Dialogue    |0.9990 |0.0002 |0.0010 |0.9990 |0.9990 |0.9990  |
> Reasoning   |1.0000 |0.0001 |0.0000 |0.9992 |1.0000 |0.9996  |
>
> ---
>
> ## 2.2 Misidentification does not harm downstream accuracy.
>
> We compare oracle routing vs. learned routing (Table.4). The results shows that the performance difference between the two settings is negligible across all benchmarks. This confirms that **routing accuracy is sufficiently high that it does not affect final task performance.**
>
>  |Method | GSM8k | MATH | HumanEval | MBPP | AlpacaEval2 | IFEval | BBH |  ARC-C |
>  |------ | ------ | ------ | ------ | ------ | ------ | ------   |------   |------   |
>  |SKillWeave (**Oracle Routing**) | 91.0 | 62.5 | 75.0 | 77.8 | 52.8 | 79.1 | 76.2 | 88.6 |
>  |SKillWeave (**Learned Routing**) | 91.0 | 62.3 | 75.0 | 77.8 | 52.9 | 78.9 | 76.2 | 88.6 |
>
>
> We manually examined all samples where the router’s prediction differs from the oracle
> domain. Interestingly, **misclassified samples often contain mixed-domain content**, such as a dialogue prompt involving embedded mathematics. We **provide case examples in Appendix B.3**: ``` You extracted 200 milliseconds of sound from a recording with a sampling rate of 48 kHz. How many amplitude measurements do you have now?’’’ ,
> where the “misrouted” “Math” skillpack is actually better suited than the oracle “Dialogue” skillpack.
>
> ---
>
> ## 2.3 Routing has negligible latency impact.
> We further measure the inference-time overhead of routing. As quantified in **Appendix C.3**, routing **increases inference time by <1%** in most settings. The small overhead is **caused by GPU memory contention** rather than routing computation itself, confirming that routing is not a bottleneck.

---

> ### Author Response · Authors · 2025-11-26
> **Rebuttal - Part II (2/3)**
>
> >  ### Critical E2E inference settings (prefill & decode length, batch size) are missing for Figure 3.
>
> We thank the reviewers for the detailed comments regarding the latency evaluation protocol.
> We have significantly expanded the latency section and **added a new Appendix C.3 “Inference Latency Measurement and Results.”** This update substantially clarifies our experimental configurations, provides standardized latency metrics, and presents new measurements across prompt lengths, generation lengths, request rates, and kernel batch sizes.
>
> ---
>
> ## 1. Comprehensive clarification of latency measurement settings
>
> Appendix C.3 now provides a complete description of our end-to-end latency setup, including:
>  - **Hardware**: all experiments on a single NVIDIA A100-80G
> - **Inference engine**: S-LoRA as the primary backend (chosen because it enables direct integration of our optimized low-rank kernels; our method remains compatible with vLLM/sGLang)
> - **Model configuration**: Llama3.1-8B-Instruct as the base model; skillpack matrices with effective rank ≈ 400 (ranging from 300 to 600 depending on the module).
> - **Request arrival process**: The requests for each domain are modeled using a Gamma arrival process with a coefficient of variance of 1Gamma arrival process (CV = 1), which produces realistic bursty workloads and is used consistently across Figures 3 and 5
> - **Reported statistics**: request throughput (req/s), token throughput (tokens/s), ms/token, and full distribution metrics (mean, median, P95, P99, min, max)
>
> These details ensure that all latency results are now fully reproducible.
>
> ---
>
> ## 2. New experiments across diverse runtime conditions (Table 6)
>
> To make latency behavior more transparent, we added **three new groups of experiments**, covering:
> 1. **Different prompt lengths** (50 / 200 / 500)
> 2. **Different generation lengths** (20 / 100 / 200 /1000)
> 3. **Different request arrival rates** (0.5 → 7 req/s)
> 4. including **long generation cases** (4000 tokens)
>
> These settings produce a broad and realistic latency landscape.
> Across all metrics, SkillWeave consistently achieves lower latency and higher throughput compared with multi-model baselines.
> We encourage reviewers to inspect Table 6 for detailed quantitative results.
>
> ---
>
> ## 3. New kernel-level latency results for different batch sizes (Table 7)
>
> Given that modern inference engines (vLLM, sGLang) use continuous dynamic batching, the actual batch size depends on concurrent traffic rather than user-specified settings. Although reviewers asked for latency under various batch sizes, end-to-end batch-size-controlled measurements are therefore not meaningful.
>
> To address this request in a principled way, we:
> - approximated effective batch size via **controlled request concurrency** (following vLLM evaluation protocols) in Table 6,
> - and added a new **kernel-level batch-sweep experiment** (Table 7) using
>     - Llama3.1-8B-Instruct
>     - down_proj matrix (4096×14336), rank=600
>     - prefill seq_len=1000, decode seq_len=1
>     - batch sizes {1, 4, 10, 32}
>
> These results clearly show that our **X8A8A8 low-rank kernel is substantially faster** than the backbone’s FP16 (X16W16) kernel across all batch settings, and the latency overhead introduced by activating a skillpack is minimal.

---

> ### Author Response · Authors · 2025-11-26
> **Rebuttal - Part III (3/3)**
>
> > ### Can you clarify on how to use the 5x7B model for domain specific tasks? Is model loading time also included in the inference latency?
>
> ## 4. Clarifying multi-model baseline deployment
>
> We thank the reviewer for asking how the 5×7B multi-model baseline is deployed and whether model-loading time is included. Appendix C.3 now provides a full, implementation-level description. For clarity, we summarize the key points below.
>
> **(a) How multiple finetuned models are hosted**
>  - Each domain-specific 7B model is run as an **independent vLLM worker** in a separate process.
>  - Nvidia **MPS** (Multi-Process Service) is used to allow multiple workers to share the same A100-80G GPU.
>  - GPU memory is **statically** divided among workers proportionally to the historical request rates of their domains.
>  - An A100-80G can host **at most 5 models** of this size; additional ones inevitably require eviction.
>
> This is the standard and most resource-efficient configuration for multi-model hosting on a single GPU.
>
> **(b) How loading/unloading works**
> - When the runtime workload shifts, we **evict the least-used model** and replace it with the model required by incoming requests.
> - Loading a new model requires **copying all parameters from CPU to the GPU** memory region associated with the corresponding vLLM worker.
> - Crucially, **CUDA Graphs are not invalidated** during this process:
>     - the vLLM execution graph remains captured,
>     - we update only the underlying weight tensors via raw memory copies,
>     - thus avoiding expensive graph re-capture.
> This process accurately reflects the best possible implementation of multi-model serving on a single GPU.
>
> **(c) Why this setting is fair**
>
> We deliberately selected the **optimal configuration N = 4** (i.e. the number of model runninng simultaneously) for multi-model baselines for the following reasons:
>  - **Too few models (e.g., N=1)**
>     - → degenerates into sequential swapping for every domain shift
>     - → catastrophically slow and not representative of real deployments
>  - **Too many models (N > 5)**
>     - → exceeds available GPU memory for 7B/8B models
>     - → vLLM workers OOM
>  - **N = 4**
>     - → no OOM
>     - → minimum number of swaps
>     - → empirically best-performing baseline on A100-80G
>
> Thus, our baseline is not artificially disadvantaged; it represents the best-case scenario for any multi-model system under realistic memory constraints.
>
> **(d) Is loading time counted?**
>
> **Yes.**
> All loading/unloading overhead—including parameter copying and the resulting queuing delays—is fully included in the end-to-end latency. This reflects real-world serving conditions, where users must wait while the server loads or activates the correct model.
>
> ---
> ---
> ---
>
> ### Once again, we sincerely appreciate the reviewer’s time and thoughtful feedback. Your comments have helped us significantly strengthen the clarity and completeness of our manuscript. If any further concerns remain, we would be more than happy to address them.

---

### Author Response · Authors · 2025-12-02
**Summary of our efforts during the rebuttal stage**

We sincerely thank the reviewers and the AC for their time and the insightful feedback. During the rebuttal period, we conducted substantial additional experiments, clarified several misunderstood components, and reorganized major portions of the paper. Below we summarize the key improvements.

---

## 1. Comprehensive clarification of routing mechanism

We added a new Appendix section “Routing Implementation” and significantly expanded routing-related experiments.

**Clarifying the original routing setup**:
- For agent-based tasks, routing is deterministic in real deployments because the tool type is explicitly known.
- For general tasks (Table 1), we now support two settings: using ground-truth domain labels (oracle) or using a lightweight domain router.

**New empirical analyses added:**
- **Routing accuracy is extremely high.  (Table 3)** Our classifier (Qwen2.5-0.5B + linear head) achieves >0.999 accuracy/precision/recall on every domain; FPR/FNR < 0.002
- **Routing errors hardly affect downstream performance. (Table 4)** Oracle routing vs. learned routing show negligible differences across all benchmarks.
- **Misrouting case studies included. (Figure 4)** We now provides representative mixed-domain examples, illustrating why occasional misclassification can still yield correct outputs.
- **Routing overhead is minimal**  Across all realistic settings, routing increases end-to-end latency by <1%, and the overhead stems primarily from GPU memory contention

These additions directly address all reviewer questions regarding routing reliability and inference impact.

---

## 2. Clarifying the meaning of “self-improvement” and addressing misunderstandings

Multiple reviewers interpreted “self-improvement” as relying solely on rule-based verification or data quantity. We substantially clarified this concept both theoretically and empirically.


**Conceptually**, SkillWeave’s self-improvement arises from:
- using **only self-generated data** (no human-written solutions, no stronger teacher), and
- reorganizing the model’s internal structure through domain-isolated full finetuning followed by compression—allowing the model to extract, refine, and **redistribute its own capabilities.**

**Empirically**, we conducted **new controlled experiments** with strict fairness settings:
- all baselines trained on identical self-generated DPO data,
- all models using the exact same size (10B),
- five strong baselines included.

Across all domains and metrics, **SkillWeave remains consistently better, demonstrating that the performance gains arise from our architecture**—full-tuning → domain decomposition → compression—not soley from privileged data or rule-based heuristics only.

---

## 3. Additional details and new experiments for latency measurement

Reviewers asked for complete transparency on latency evaluation. We added an extensive new section (Appendix C.3) with the full methodology.

- **Fully detailed E2E latency setup (Appendix C.3)**. We now specify hardware, inference engine, model configuration, request arrival distribution, and all latency metrics reported.
- **Three new groups of experiments (Table 6)**. We systematically vary prompt lengths (50/200/500), generation lengths (20/100/200/1000/4000) and request arrival rates (0.5 → 7 req/s). These results expose the runtime behavior across diverse real-world conditions.
- **New kernel-level latency sweep (Table 7)**.  Across batch sizes {1,4,10,32}, our X8A8A8 low-rank GEMM kernel is consistently faster.
- **Clarification of the 5×7B multi-model baseline**. We added a transparent description of how the multi-model baseline is deployed (vLLM + MPS, memory-aware scheduling), making Figure 3 and 5 fully reproducible.

---

## 4. New evidence on interference and forgetting

To directly evaluate whether SkillWeave alleviates cross-domain interference and catastrophic forgetting, we conducted two complementary analyses:
- **Sequential training experiment.**
We trained models sequentially across domains and measured performance decay. Mixed-domain and continual-learning baselines exhibit clear degradation, while SkillWeave maintains stable accuracy, across the entire sequence, indicating minimal forgetting.

- **Parameter-level conflict analysis.**
We further analyzed the learned task vectors and the merged backbone:
    - **Conflict analysis**: task vectors across domains show low cosine similarity and negative sign consistency, confirming that full multi-domain finetuning naturally induces interference.
    - **Shared-knowledge extraction**: the merged backbone shows high similarity with all task vectors, and post-merge vectors exhibit reduced magnitude (~33% decrease), demonstrating that merging isolates shared capabilities and makes compression more stable.

Together, these results reinforce that SkillWeave systematically mitigates task interference and forgetting.

---

### Note · Authors · 2025-12-31

I have read and agree with the venue's withdrawal policy on behalf of myself and my co-authors.